# Detection of collapsed buildings due to the 2016 Kumamoto, Japan, earthquake from LiDAR data

Luis Moya[1], Fumio Yamazaki[2], Wen Liu[2], Masumi Yamada[3]

[1]International Research Institute of Disaster Science, Tohoku University, Miyagi, Sendai, 980-0845, Japan
[2] Department of Urban Environment Systems, Chiba University, Chiba 263-8522, Japan
[3]Disaster Prevention Research Institute, Kyoto University, Gokasho, Uji, 611-0011, Japan

*Correspondence to*: Luis Moya (lmoyah@irides.tohoku.ac.jp)

**Abstract.** The 2016 Kumamoto earthquake sequence was triggered by an Mw 6.2 event at 21:26 on April 14. Approximately 28 hours later, at 1:25 on April 16, an Mw 7.0 event (the mainshock) followed. The epicenters of both events were located

near the residential area of Mashiki town and the region nearby. Due to very strong seismic ground motion, the earthquake produced extensive damage to buildings and infrastructure. In this paper, collapsed buildings were detected using a pair of digital surface models (DSMs), taken before and after the April 16 mainshock by airborne light detection and ranging (LiDAR) flights. Different methods were evaluated to extract collapsed buildings from the DSMs. The change of the average elevation within a building footprint was found to be the most important factor. Finally, the distribution of collapsed

buildings in the study area was presented, and the result was consistent with that of a building damage survey performed after the earthquake.

## 1 Introduction

The detection of affected areas after an earthquake is very important for disaster response activities. Allocating resources, such as relief forces, food, medicine and shelter, is crucial after a natural disaster strikes (Das and Hanaoka, 2014). Thus,

proper information on the damage situation will improve the efficiency in distributing relief resources. The extent of the affected area also provides an idea of the scale of the disaster and an estimate of the relief demand. Damage assessment after an earthquake disaster is important for the scientific community as well. A significant amount of information has been obtained from previous earthquakes and used to improve construction design codes to evaluate and mitigate damage to buildings and infrastructure in the event of future earthquakes. For instance, Whitman et al. (1973) provided earthquake

damage probability matrices using data collected after the 1971 San Fernando, California earthquake. Yamazaki and Murao (2000) proposed vulnerability functions for Japanese buildings based on building inventory and damage data and the spatial distribution of strong motion (Yamaguchi and Yamazaki, 2001) during the 1995 Kobe, Japan, earthquake.

Information gathered from field surveys is invaluable and very precise; however, the process requires significant time and effort, and access to affected areas is often hindered by road closures and secondary hazards. Remote sensing is an effective

tool for detecting damaged areas because it can be used to document damage to large areas without direct access to the affected area (Yamazaki and Matsuoka, 2007; Rathje and Adams, 2008; Dell'Acqua and Gamba, 2012). Immense improvement to the accessibility of remote sensing imagery data and geospatial data processing tools has been achieved over the last several years (Vuolo et al., 2016; Korosov et al., 2016). A dramatic increase in the number of satellite, aircraft, and

unmanned aerial vehicle (UAV) sensors has been observed as well. One of the most successful approaches for assessing damaged areas is based on change detection between a pair of images taken before and after an earthquake (Meslem et al., 2011; Liu et al. 2013; Uprety et al., 2013). In addition, remote sensing has been used for long term urban recovery monitoring (Hoshi et al., 2014; Hashemi-Parast et al., 2016).

    Schweier and Markus (2006) pointed out that airborne light detection and ranging (LiDAR) data can be used to classify
collapsed buildings using the following geometrical features of a building extracted from LiDAR data: the height change from the initial one, the reduction of the total volume, the footprint borders, the inclination of the structure, the debris spread outside the footprint, the additional covered area outside the footprint, and the damage situation of the roof. They proposed a modification of the previous damage classification method (Okada and Takai, 2000) using these geometrical features. Although they suggested the use of airborne LiDAR data in extraction of collapsed buildings, applications to real cases were
not provided.

    Applications of LiDAR for damage detection are still few compared with other remote sensing technologies. The main reason is the lack of LiDAR data before a disaster. However, Aixia et al. (2016) performed a study on the possibility of detecting building damage using only a post-earthquake LiDAR digital surface model (DSM). Their results are promising for buildings with simple roof shapes, such as flat and pitched roofs. Rehor et al. (2008) proposed the use of a planes-based
segmentation method to detect damaged buildings, where the number of unsegmented pixels in damaged buildings is larger than in undamaged buildings. Labiak et al. (2011) proposed an automated method to detect and quantify building damage using only a post-earthquake LiDAR DSM as well, but their results had low accuracy for heavily damaged and collapsed buildings. Hussain et al. (2011) combined LiDAR data with GeoEye-1 imagery to detect damaged buildings after the 2010 Haiti earthquake. They detected 190 damaged buildings out of 200; however, their procedure required manual intervention,
and the damage level was not clearly classified. Instead of LiDAR data, Maruyama et al. (2014) constructed two digital surface models (DSMs) from two sets of aerial images: before and after the earthquake. Then, the collapsed buildings after the 2007 Niigata-Chuetsu-Oki, Japan, earthquake were extracted using the difference of elevation between the DSMs.

    An $M_w$ 6.2 earthquake struck Kumamoto Prefecture, Japan, on April 14, 2016 at 21:26 JST. The event produced structural damage and resulted in nine human casualties (Cabinet Office of Japan, 2016). Then, 28 hours later, a second earthquake
with $M_w$ 7.0 occurred close to the first one. Thus, the first event was designated the "foreshock" and the second the "mainshock". The epicenter of the foreshock was located at the end of the Hinagu fault, and the epicenter of the mainshock was located in the Futugawa fault. Both events were located in Mashiki town with a population of 33 thousand. The number of aftershocks following these events reached the largest number among recent inland earthquakes in Japan (Japan

Meteorological Agency, 2017). The total number of deaths due to direct causes reached fifty, and over eight-thousand residential buildings were severely damaged or collapsed due to the Kumamoto earthquake sequence.

Among the several remote sensing technologies used to monitor the area affected by the Kumamoto earthquake (Yamazaki and Liu, 2016), a pair of LiDAR datasets taken before and after the mainshock were available (Moya et al., 2017).
As mentioned before, this kind of dataset is not often available. Therefore, this study explores the potential use of LiDAR data to extract collapsed buildings over the affected area. Building collapse is still the main cause of casualties and hence its prompt recognition is crucial for search and rescue operations. The difference of elevation, the standard deviation and the correlation coefficient are tested in this for this purpose.

## 2 Study area and dataset

After the foreshock, a LiDAR surveying flight was carried out during 15:00 – 17:00 (JST) on April 15, 2016, in order to record the effects of the earthquake (Asia Air Survey Co., Ltd., 2016). It produced point clouds with an average point density of 1.5-2 points/m$^2$. Subsequently, because the unexpected mainshock occurred, a second mission was set up during 10:00 – 12:00 (JST) on April 23, 2016, which produced point clouds with an average point density of 3-4 points/m$^2$. Both sets of LiDAR data were acquired using a Leica ALS50II instrument and the same pilot and airplane. After rasterization of the raw
point clouds, two digital surface models (DSMs) with a data spacing of 50 cm were created. The DSM collected before and after the mainshock will hereafter be referred to as the BDSM and ADSM, respectively. Figure 1 shows the extent of the ADSM, which represents the entire study area. It covers the main part of Mashiki town and some parts of Nishihara village, Mifune and Kashima towns, and Kumamoto city.

The study area is located in the near field of the Kumamoto earthquake sequence where significant permanent ground
displacements were produced during the earthquake. A direct comparison of the BDSM and ADSM shows that the building coordinates do not match because the ADSM contains coseismic displacements. Therefore, the ADSM was shifted before detecting the damaged buildings based on the permanent crustal movement calculated by Moya et al. (2017). To do this, an automated procedure for calculating the permanent three-dimensional (3D) displacement was implemented. The permanent ground displacement was calculated by 100m-grid size, and then it is applied to the ADSM pixels within the grid-size.
Figure 2 illustrates the calculated permanent ground displacement of the common LiDAR data area. In the figure, the results of new field measurement carried out in August 2016 for surveying reference points after the Kumamoto earthquake are also shown (Geospatial Information Authority of Japan, 2017). The coseismic displacements estimated from the LiDAR data show good agreement with the survey results (Figure 3). In Figure 2, the causative fault is located in the areas where sudden changes in the direction of the permanent ground displacement are observed. Over the entire study area, a maximum
horizontal displacement of approximately 2 m was observed.

## 3 Detection of damaged buildings

To focus on buildings, a geocoded building footprint dataset, provided by the Geospatial Information Authority of Japan (GSI), was used. Only buildings with footprint areas greater than 20 m² were evaluated. Because the point densities of the BDSM and ADSM are different and the footprint data include some errors, perfect matching of the DSMs with the building footprints could not be achieved. For this reason, the building footprints were reduced by 1 m (i.e., the reduced polygon is located inside a building footprint), and they were projected onto the same reference system as that of the DSMs (Figure 4). The LiDAR data within the reduced building boundaries were then extracted and processed. The reason for using the reduced building boundaries was to discard the DSM data near the building's boundaries in the subsequent analysis. The distance of the buffer (1 m) was decided based on a preliminary evaluation of the data (Moya et al., 2016).

Figure 4 illustrates five buildings located in the study area. For each case, the BDSM (blue dots), the ADSM (red dots), and the difference of the two DSMs are depicted. These buildings were selected in order to demonstrate different damage patterns: non-damaged, tilted, and collapsed buildings. It is worth noting that the difference between the DSMs for a non-damaged building (Figure 4a) shows high values around the boundary of the building footprint, which was caused by the effect mentioned earlier. These errors are certainly present for tilted buildings as well and make damage detection very challenging (Figure 4b). Figure 4c shows a typical collapsed steel-frame building with a well-known damage pattern that occurs with a *soft story* or a *weak story*, that is, a significant difference of the stiffness/resistance between one story and the rest. They show a significant horizontal/vertical movement, which is easier to detect by LiDAR data. Figure 4d shows a collapsed wooden building that was shifted significantly in the horizontal direction; conversely, the collapsed wooden building shown in Figure 4e does not exhibit such horizontal movement, only a vertical shift. Lateral spread of debris is an important issue when the building is located along a main road. For almost all the collapsed buildings, a clear decrease in building elevation was observed from the LiDAR DSMs.

The number of buildings within the study area is very large, so it is necessary to implement an automated procedure to evaluate the extent of their damage. In this study, three parameters were used: the average height difference between the two DSMs ($\Delta H$) within the reduced building footprint, its standard deviation ($\sigma$), and the correlation coefficient ($r$) between the two DSMs. These parameters were calculated for each building using the following equations:

$$\Delta H = \frac{1}{N} \sum_{i=1}^{N} (Ha_i - Hb_i) \tag{1}$$

$$\sigma = \sqrt{\frac{\sum_{i=1}^{N}\left((Ha_i - Hb_i) - \Delta H\right)^2}{N}} \tag{2}$$

$$r = \frac{N \sum_{i=1}^{N} Ha_i Hb_i - \sum_{i=1}^{N} Ha_i \sum_{i=1}^{N} Hb_i}{\sqrt{\left(N \sum_{i=1}^{N} Ha_i^2 - \left(\sum_{i=1}^{N} Ha_i\right)^2\right)\left(N \sum_{i=1}^{N} Hb_i^2 - \left(\sum_{i=1}^{N} Hb_i\right)^2\right)}} \tag{3}$$

where $i \in \{1, 2, …, N\}$ and $N$ is the number of elevation points inside a given reduced building footprint. $Ha_i$ and $Hb_i$ are the elevations from the ADSM and BDSM, respectively. The correlation coefficient ranges from -1.0 to 1.0 and has proven to be effective in detecting changes from a pair of satellite images (Liu et al., 2013; Uprety et al., 2013). A value of $r$ close to 1.0 indicates no change.

Yamada et al. (2017) presented the distribution of building damage in the central part of Mashiki, where the damage was determined from aerial photos and field surveys. The damaged buildings were classified into four categories: no damage (D0), partially/moderately damaged (D1-D3), severely damaged/inclined (D4) and story collapse (D5). Here, D1-D5 represents the degree of damage according to Okada and Takai (2000), which is similar to G1-G5 of the European Macroseismic Scale (EMS-98). Figure 5 shows the damage distribution over the surveyed area, which is located along the north side of the Akitsu River.

Figure 6 shows the scatter plots of the parameters calculated for the surveyed buildings, and Figure 7 shows the histograms of the three parameters for the buildings with different damage levels, where the average (solid line) and the standard deviation (dashed line) are also included. Significant overlap of damage levels D0, D1-D3, and D4 was observed regardless of which parameter was chosen. On the other hand, collapsed buildings (D5) tend to have large negative values of $\Delta H$. Therefore, this paper focuses on the detection of collapsed buildings. It is important to note that few collapsed buildings show positive values of $\Delta H$. A closer look showed that those buildings were covered by a neighboring building that had collapsed.

Although $\Delta H$ seems to be the dominant parameter for extracting collapsed buildings, the other two parameters ($\sigma$ and $r$) can still provide additional information. For instance, if we observe the collapsed buildings from the scatter plot in Figure 6c (red marks), a trend can be observed in which $r$ trends to one when $\sigma$ is close to zero. This trend is related to the collapse patterns and can be observed for the collapsed buildings shown in Figure 4. Figure 4d shows a completely collapsed building, where the debris has spread laterally. For those cases, the values of $r$ are low and the values of $\sigma$ are large. On the other hand, Figure 4e shows a collapsed building whose roof remained almost the same shape while it collapsed almost vertically. This means that all the elevations inside the footprint decreased by about the same amount, thus leading to a high value of $r$ and a low value of $\sigma$. This pattern is often difficult to detect from optical aerial and satellite optical images because the sensor measures the landscape vertically. The histograms for collapsed buildings (Figure 7) shows that several collapsed buildings have a value for $r$ greater than 0.5, and it would be difficult to detect this from aerial or optical satellite imagery. Readers might notice that non-collapsed buildings also have a value of $r$ close to 1 and $\sigma$ close to zero; however, those can be first filtered using $\Delta H$. Then, the pattern of collapse can be evaluated from the other two parameters (a process known as a decision tree).

Within the study area, 26,128 building footprints were extracted. It is worth mentioning that few buildings were not well registered in the GIS map. Figure 8 shows the parameters calculated for each building, where the shaded color depicts the density of the dots. As seen, most of the points are located at approximately $(\Delta H, \sigma, r) = (0\ m, 0.5\ m, 0.9)$, which indeed

represents non-collapsed buildings. Several buildings show positive values of $\Delta H$. A closer look revealed two principle factors: (1) the collapse of a neighboring building and (2) plastic covers placed over the roof for protection from the rain.

The next concern was to define a criterion to set threshold values that can differentiate collapsed/non-collapsed buildings properly. A number of options were evaluated in this study. Since it is obvious that the buildings with clear negative values of $\Delta H$ correspond to collapsed buildings, we first analyzed the classification using a threshold for $\Delta H$ only. The buildings whose $\Delta H$ values were smaller than that threshold were classified as collapsed; in contrast, the buildings whose $\Delta H$ values were greater than the threshold were classified as non-collapsed. The possible thresholds were tested on the buildings surveyed by Yamada et al. (2017). Figure 9 shows the Cohen's Kappa coefficient and the overall accuracy calculated from the comparison between the estimated collapsed and non-collapsed buildings (i.e., using a given threshold) and the building damage classes based on the ground truth. For the comparison, the buildings with damage levels D0, D1-D3, and D4 were labelled non-collapsed buildings. The Cohen's Kappa ($k$) coefficient and the overall accuracy (OA) are expressed as follows:

$$p_{no} = (p_{21} + p_{22})(p_{12} + p_{22}) \tag{4}$$

$$p_{yes} = (p_{11} + p_{12})(p_{11} + p_{21}) \tag{5}$$

$$p_o = p_{11} + p_{22} \tag{6}$$

$$p_e = p_{yes} + p_{no} \tag{7}$$

$$OA = p_o \tag{8}$$

$$k = \frac{p_o - p_e}{1 - p_e} \tag{9}$$

where $p_{11}$ and $p_{12}$ are the ratio of non-collapsed buildings predicted as non-collapsed and collapsed buildings, respectively. $p_{21}$ and $p_{22}$ are the ratio of collapsed buildings predicted as non-collapsed and collapsed buildings, respectively. From Figure 9, it is observed that a threshold value of -0.5 m gave the highest values for both the Cohen's Kappa coefficient (0.80) and the overall accuracy (0.93).

To determine if the use of all the parameters could produce better accuracy in detecting collapsed buildings, the Support Vector Machine (SVM) method was selected to construct a plane that separates collapsed and non-collapsed buildings in the three-dimensional database ($\Delta H$, $\sigma$, $r$). The plane has the largest distance from the nearest training data (ground truth data). Using kernel functions, SVM can be used to construct a non-linear function as well; however, in this study we only evaluated linear functions (i.e., a plane or linear kernel function). Figure 10 shows the plane constructed using SVM, where the red and blue marks depict the collapsed and non-collapsed buildings based on the ground truth, respectively. The plane was constructed using the same amount of data for the two classes. Thus, 205 non-collapsed buildings were selected randomly from the surveyed data. The analysis was performed several times, and although the plane obtained showed small variations due to the random selection of the training data, the Cohen's Kappa coefficient produced in each analysis was almost constant with minor fluctuations around 0.80. The accuracy produced by SVM is very similar to the accuracy

obtained when only a $\Delta H$ threshold is used. For a linear kernel SVM, the vector $\mathbf{w}$ perpendicular to the decision plane is defined by the following expression:

$$\mathbf{w} = \sum_i \alpha_i y_i \mathbf{x}_i \qquad (10)$$

where $\mathbf{x}_i$ is a training vector that contains the three parameters ($\Delta H$, $\sigma$ and $r$), $y_i$ represent the class that can be either 1 or -1, and the coefficients $\alpha_i$ are obtained by solving the following problem:

$$\min_{\boldsymbol{\alpha}} \left( \frac{1}{2} \boldsymbol{\alpha}^T \mathbf{Q} \boldsymbol{\alpha} - \mathbf{e}^T \boldsymbol{\alpha} \right) \qquad (11)$$

$$Q_{ij} = y_i y_j \left( \mathbf{x}_i \cdot \mathbf{x}_j \right) \qquad (12)$$

$$0 \le \alpha_i \le C, i = 1,...,n \qquad (13)$$

where $e$ is a vector whose elements are all ones, C is the upper bound and is used as a regularization parameter.

     The parameter C trades off misclassification of training examples against simplicity of the decision surface. A low C value makes the decision surface smooth and a high C value aims at classifying all training examples correctly (Skit-learn, 2017a). In this study a value C equals to 1 was used. In order to evaluate its effects, a cross-validation procedure was performed. For each C value, 80% of the surveyed data were selected randomly and were used to calibrate the SVM classifier. The rest of

the surveyed data were used to calculate a score that represents the accuracy. The overall accuracy was chose as the score. The procedure was repeated 5 times and the average was stored. Figure 11 shows the cross-validation accuracy. It is observed the accuracy remains mainly constant with small fluctuations at lower values. However, a difference of approximately 3% is observed between the worst and the best accuracy. Therefore, it is concluded that the C value did not affect the SVM classifier in our study.

This study also evaluated the potential use of unsupervised classification to extract collapsed buildings. Specifically, K-means cluster analysis was applied to all the data in the study area. Unlike SVM, K-means clustering does not require training data. Therefore, the database of all the buildings in the study area (Figure 8) was used. The method clusters the data and separates them into two groups, which represent the collapsed and non-collapsed buildings. The objective of the method is to minimize the inertia of each group, that is, the summation of the squared distance between all the data points of a group

and its centroid. The result is highly dependent on the initialization of the centroids. Here, k-means++ initialization scheme was used. K-means++ initializes the centroids to be distant from each other (Scikit-learn, 2017b). Figure 12 represents the predicted collapsed and non-collapsed buildings using the K-means clustering method, for which the Cohen's Kappa coefficient obtained was 0.76. Figure 13 shows the confusion matrix calculated from the comparison between the ground truth data and the predicted results from the three methods explained above: applying a $\Delta H$ threshold, SVM, and K-means

clustering. The first two methods show the same level of accuracy, while K-means clustering shows a lower accuracy.

     Figure 14 illustrates the spatial distribution of collapsed buildings estimated using a $\Delta H$ threshold of -0.5 m. A large number of collapsed buildings were observed in the study area (Figure 14a). The red and black polygons represent the collapsed (D5) and non-collapsed (D0-D4) buildings, respectively. The color of the pixels represents the difference in

elevations between the ADSM and BDSM. Blue pixels depict differences of elevations less than -0.5 m, and yellow pixels represent differences greater than 0 m. Figure 14b and Figure 15 provide a closer look of the areas where the collapsed buildings are concentrated. Figure 14b also depicts the location of the collapsed buildings surveyed by Yamada et al. (2017) as black triangles. Within the study area, a total of 26,128 buildings were evaluated, and 1,760 buildings were classified as
collapsed ($\Delta H$ less than -0.5 m).

     It was observed that some buildings collapsed by the foreshock (April 14 event) were also detected by the LiDAR methods. In order to be detected, the debris of those buildings should be either severely disturbed by the mainshock (April 16 event) or removed before the ADSM was recorded. For instance, Figure 16 shows two buildings collapsed during the foreshock (Figure 16a). However, because the mainshock produced significant reduction of their elevations (Figure 16b), it
was also detected from the pair of LiDAR.

## 4 Discussion

This paper evaluated the use of LiDAR data to detect damaged buildings by means of three parameters: $\Delta H$, $\sigma$, and $r$. It was found that collapsed buildings can be extracted precisely from the average difference of heights, $\Delta H$. However, the other two parameters can provide additional information about the collapse pattern. The collapsed patterns are correlated to the failure
mechanism of buildings, which might highlight some deficiencies of the design codes that was used in the construction process. A detailed understanding of the failure mechanism is important to the practice of forensic engineering, the investigation of failures and other performance problems. Moreover, with further evaluation, the collapsed pattern might contribute to future improvements of the building design codes. Unfortunately, it was not possible to calibrate a threshold that can properly classify different collapsed patterns. The main reason is because there was no information related to the
collapse patterns in the survey data. Perhaps this task can be done in a further research after new survey data are released.

     Some words regarding sources of error that were present in this study should be mentioned. The footprint data provided by the Geospatial Information Authority of Japan (GSI) is rather precise but not perfect. Three drawbacks were observed: (1) a few buildings were not included in the database, (2) a slight shift between the building footprint and corresponding LiDAR data is sometimes observed, and (3) in some cases, a group of buildings, consisting mostly of two or three buildings, were
registered within one building footprint. These uncertainties may have produced errors in the detection of collapsed buildings. However, they did not have a significant impact on the overall results, which is confirmed in Figure 9, where the Cohen's Kappa coefficient and the overall accuracy are significantly high. This problem can be solved by performing manual inspection or automatic detection of buildings from the BDSM in order to update the dataset. However, the authors decided to work with the data in its current state because this uncertainty is likely to be present in other real situations where
a quick report on damage extent is required.

     Of the three methods evaluated here, the K-means clustering exhibits the lowest accuracy. The main reason is that, unlike the SVM method, the K-means clustering does not use any truth data. However, it produced Kappa coefficient of 0.76 and

overall accuracy of 92%, which is still quite good. The K-means clustering method is useful for taking a first glance at the distribution of collapsed buildings because the method does not require any training data. The procedure is well-known and robust, with several efficient algorithms with proven fast convergence.

Finally we would like to discuss on the building damage rate in the Kumamoto earthquake, compared with other recent M7-level crustal earthquakes in Japan. In the 1995 Kobe earthquake (Mw 6.9), about 49 thousand buildings were collapsed (G5 in EMS-98 scale) or severely damaged (G4) out of 560 thousand buildings in the affected urban area (Building Research Institute, 1996). The recorded strong motion distribution in the Kobe earthquake was in the similar level as that of the Kumamoto earthquake (Yamaguchi and Yamazaki, 2001). But in 1995, the number of strong motion accelerometers was much less, about only ten in the hard-hit zone. The built-up density of the affected area was much higher in the Kobe region than that of the Kumamoto region. Considering these differences, it is difficult to conclude which earthquake was more destructive. From our experiences (Yamazaki and Murao, 2000; Yamaguchi and Yamazaki, 2000), the severity of damage for wood-frame houses in Mashiki town was in the similar level as that of the hard-hit zone of the Kobe region. There were a few more recent M7-level crustal earthquakes in Japan, such as the 2004 Niigata-Chuetsu earthquake (Mw 6.6) and the 2007 Niigata-Chuetsu-Oki earthquake (Mw 6.6). But the population density of the affected urban areas was fewer in these events, and thus it is again difficult to compare their damage situations (Nagao et al., 2011) with that of Kumamoto although their strong motion levels were again comparable with that of the Kumamoto event.

Another point of discussion is whether or not the Mw 6.2 foreshock influenced the overall damage situation of Mashiki town. Our preliminary conclusion is "partially yes, but not so much". Based on numerical analyses on the collapse behaviour of typical wood-frame buildings (Building Research Institute, 2015), those built by the old seismic-code (before 1982) were mostly collapsed only by the main-shock's excitation, even without pre-shaken by the foreshock (Suto et al., 2017). But in some cases, building models were collapsed only under the sequence of the foreshock and main-shock excitations. More detailed results on this matter will be presented in the near future.

## 5 Conclusions

In this study, the spatial distribution of collapsed buildings was extracted from a pair of LiDAR datasets taken before and after the 2016 $M_w$ 7.0 Kumamoto earthquake. For this purpose, geographic information of building footprints was employed. Three parameters were used: the average ($\Delta H$) and standard deviation ($\sigma$) of the height differences between the two DSMs and the correlation coefficient between them ($r$). The parameters were evaluated using the building damage survey dataset obtained by Yamada et al. (2017); $\Delta H$ was found to be very efficient for extracting collapsed buildings. However, the other parameters provided insights into the collapse pattern. After evaluating different methodologies to extract collapsed buildings, buildings with $\Delta H$ less than -0.5 m were considered as collapsed. The distribution of collapsed buildings obtained by Yamada et al. (2017) was illustrated together with the height difference between the two DSMs, and good agreement was observed. From a total of 26,128 evaluated buildings, 1,760 collapsed buildings were extracted. To our knowledge, this

result may be the first case where a large number of collapsed buildings were extracted from a pre- and post-event LiDAR DSM pair.

It is expected that the use of LiDAR data to extract damage areas will eventually increase in the near future. However, because of the current lack of data, the implementation of a method to extract of collapsed building using only post-event LiDAR data is important and will be considered in a future study. Additional future studies related to the use of these LiDAR data are the quantification of debris expansion that is related with road blockage, and the extraction of landslides.

## 6 Data and Resources

The digital surface models used in this study are owned and provided by Asia Air Survey Co., Ltd. The building footprint data are available from the web site of the Geospatial Information Authority of Japan.

## 7 Acknowledgments

This study was financially supported by a Grant-in-Aid for Scientific Research (Project numbers: 17H02066, 24241059) and the Core Research for Evolutional Science and Technology (CREST) program by the Japan Science and Technology Agency (JST) "Establishing the most advanced disaster reduction management system by fusion of real-time disaster simulation and big data assimilation (Research Director: Prof. S. Koshimura of Tohoku University)".

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

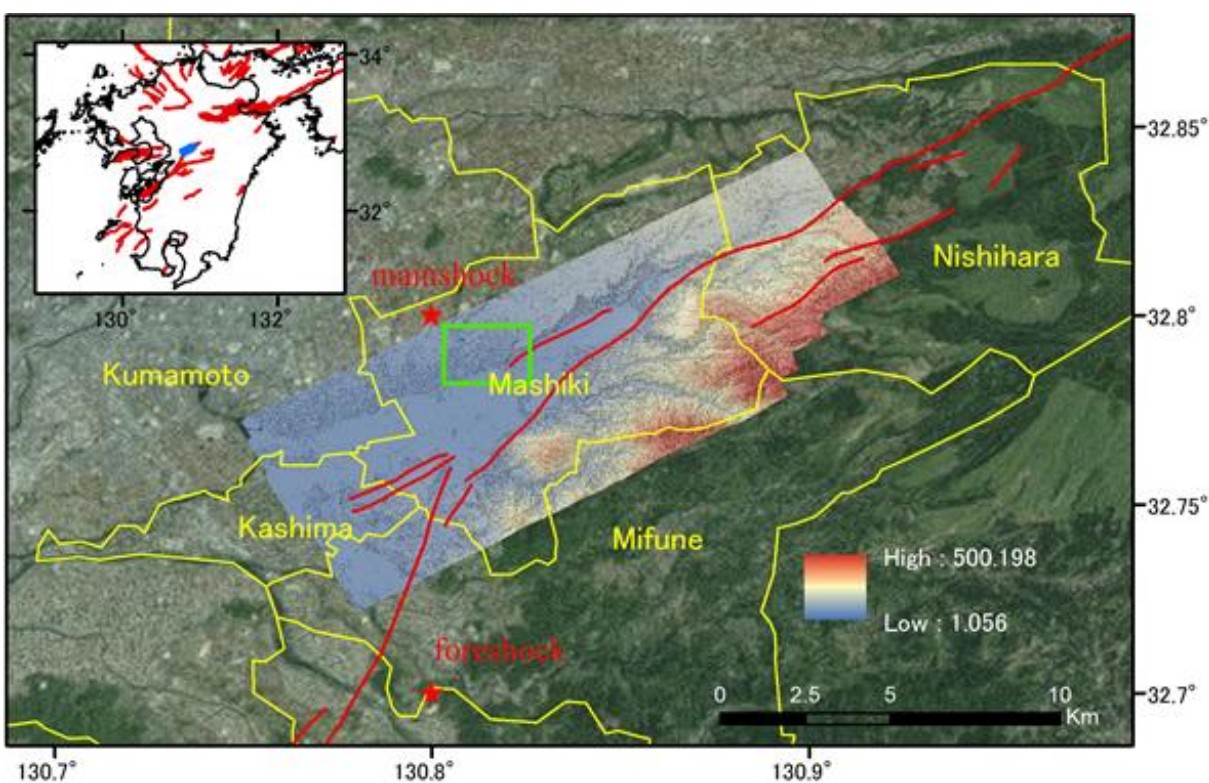

**Figure 1: The post-event LiDAR data for the study area. Shaded colors represents the elevation. The green rectangle shows the locations of the area surveyed by Yamada et al. (2017). The inset shows Kyushu Island, and the blue polygon in the inset depicts the study area**

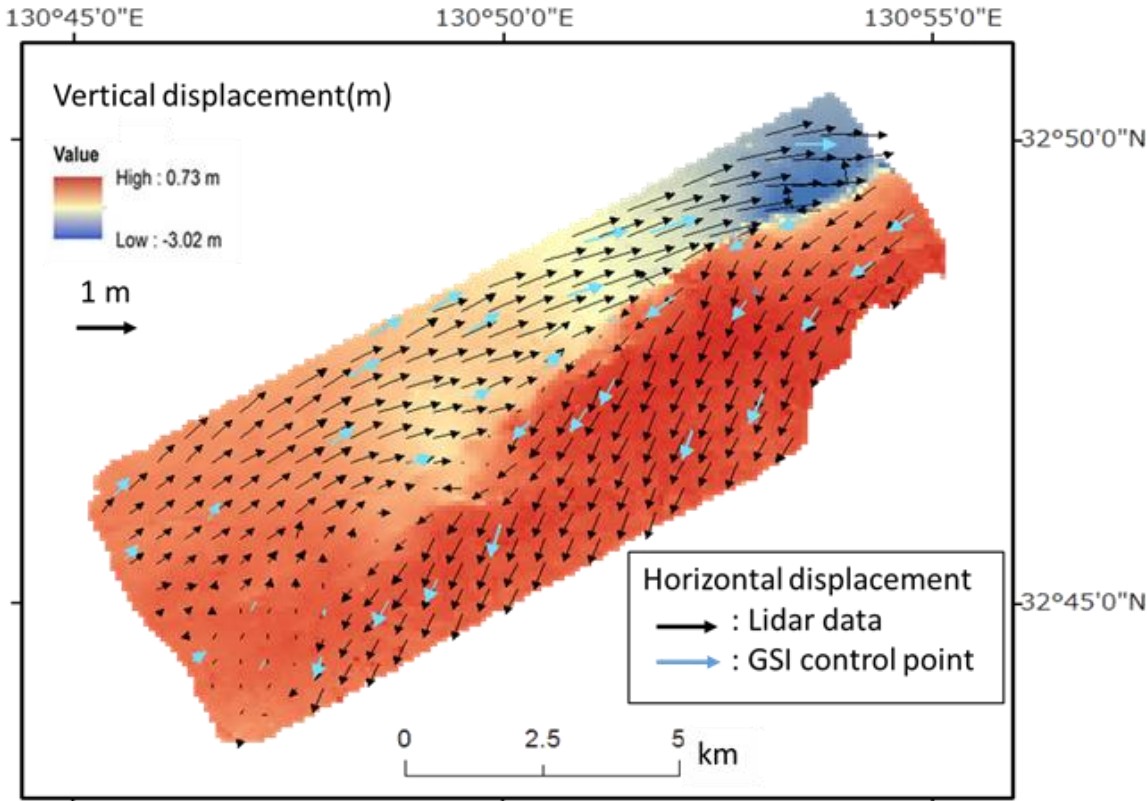

**Figure 2: Estimated three-dimensional coseismic displacement after the mainshock of the 2016 Kumamoto earthquake. The black arrows and the shaded colors indicate the horizontal and vertical displacements obtained from LiDAR, respectively (Moya et al., 2017). The blue arrows indicate the horizontal displacements at the control points measured by the Geospatial Information Authority of Japan (2016).**

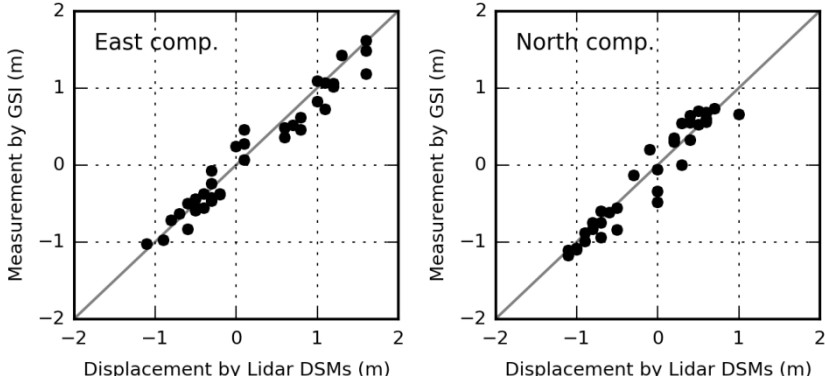

**Figure 3: Comparison between the coseismic displacements estimated from the LiDAR data (Moya et al. 2017) and from field measurements (Geospatial Information Authority of Japan, 2016)**

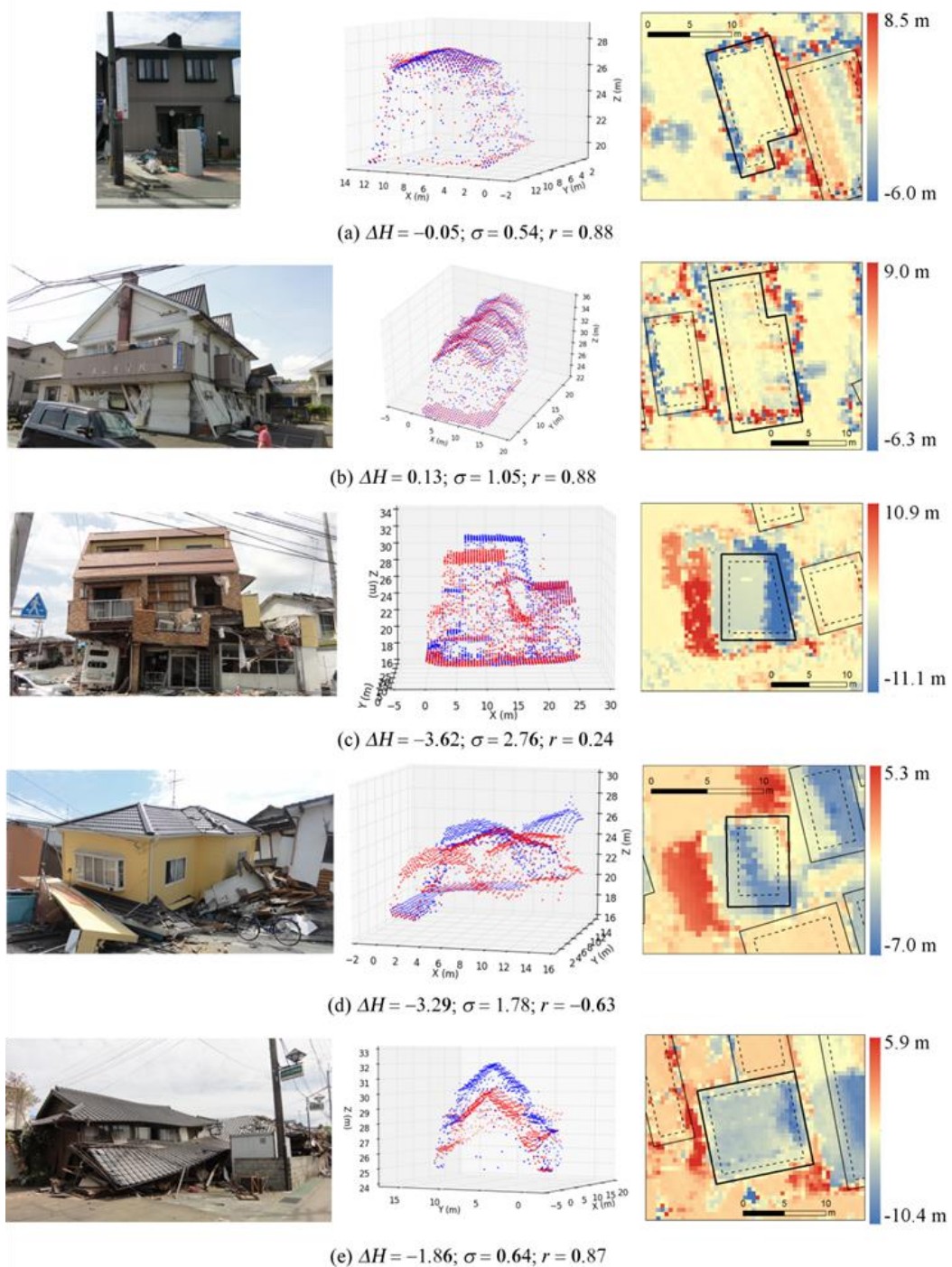

(a) $\Delta H = -0.05$; $\sigma = 0.54$; $r = 0.88$

(b) $\Delta H = 0.13$; $\sigma = 1.05$; $r = 0.88$

(c) $\Delta H = -3.62$; $\sigma = 2.76$; $r = 0.24$

(d) $\Delta H = -3.29$; $\sigma = 1.78$; $r = -0.63$

(e) $\Delta H = -1.86$; $\sigma = 0.64$; $r = 0.87$

**Figure 4: Examples of collapsed buildings extracted using the LiDAR data. The left column shows the photos taken after the mainshock by the authors. The middle column shows the LiDAR data, where the blue points depict the BDSM and the red points the ADSM. The right column shows the elevation differences between the two DSMs, where the solid lines depict the building footprints and the dashed lines depict the footprint reduced by 1m.**

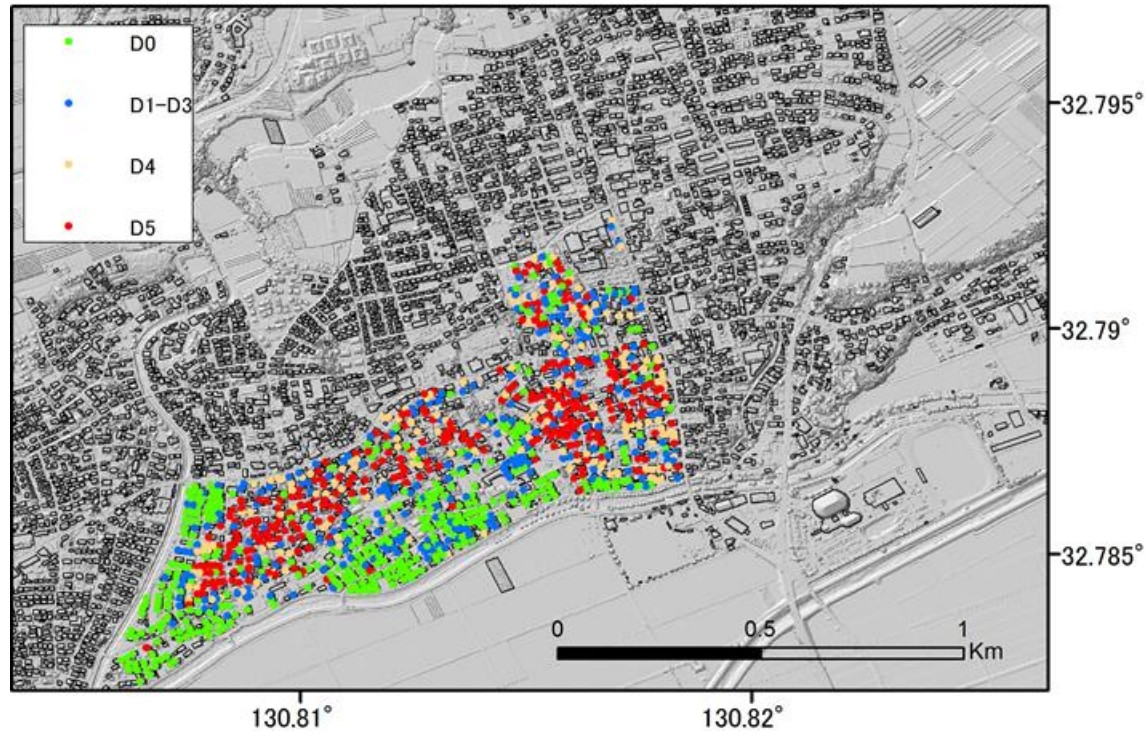

**Figure 5: Building damage survey data from Yamada et al. (2017). The location of the survey area is shown in Figure 1.**

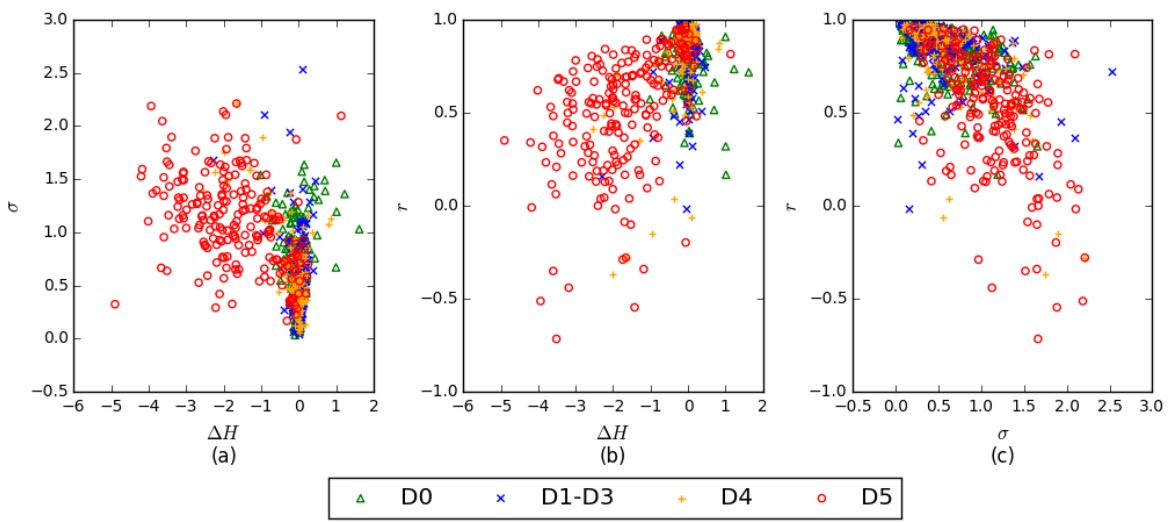

**Figure 6: Scatter plots of the three parameters ($\Delta H$, $\sigma$, $r$) calculated from the LiDAR DSMs for the buildings surveyed by Yamada et al. (2017).**

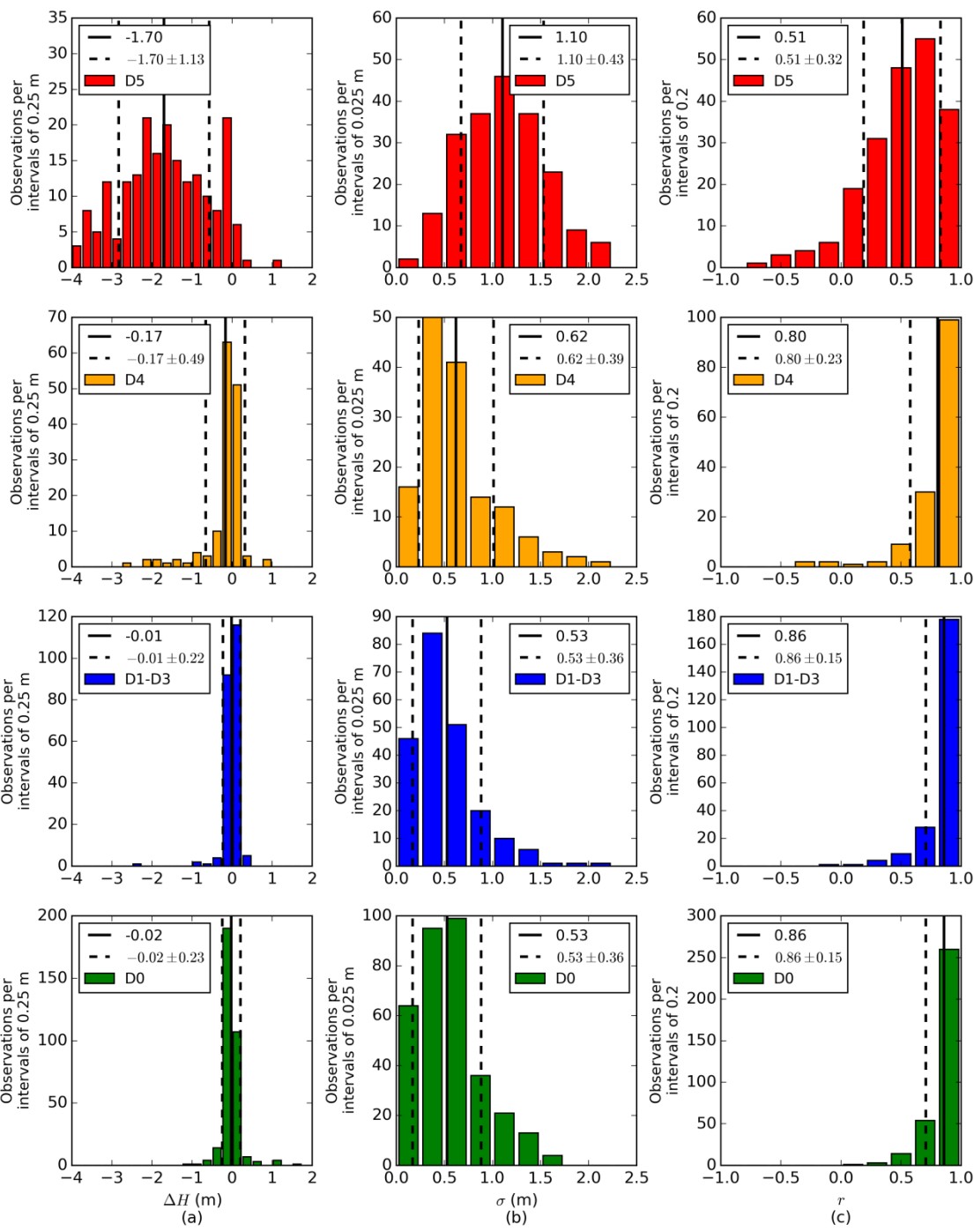

**Figure 7: Histograms of the three parameters ($\Delta H$, $\sigma$, $r$) calculated from the LiDAR DSMs for the buildings surveyed by Yamada et al. (2017), separated into four damage levels.**

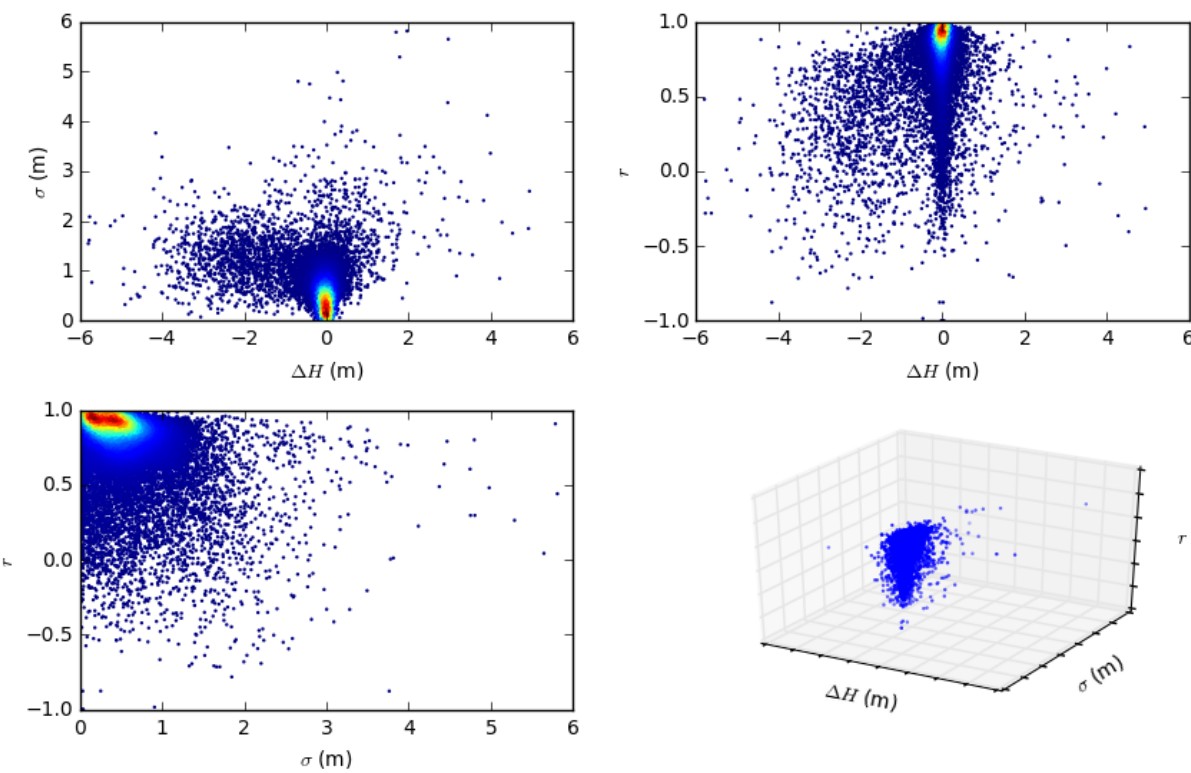

**Figure 8: Scatter plots of the three parameters ($\Delta H$, $\sigma$, $r$) calculated for all the buildings in the study area. The color represents the density of points, where red shows the area with the highest density.**

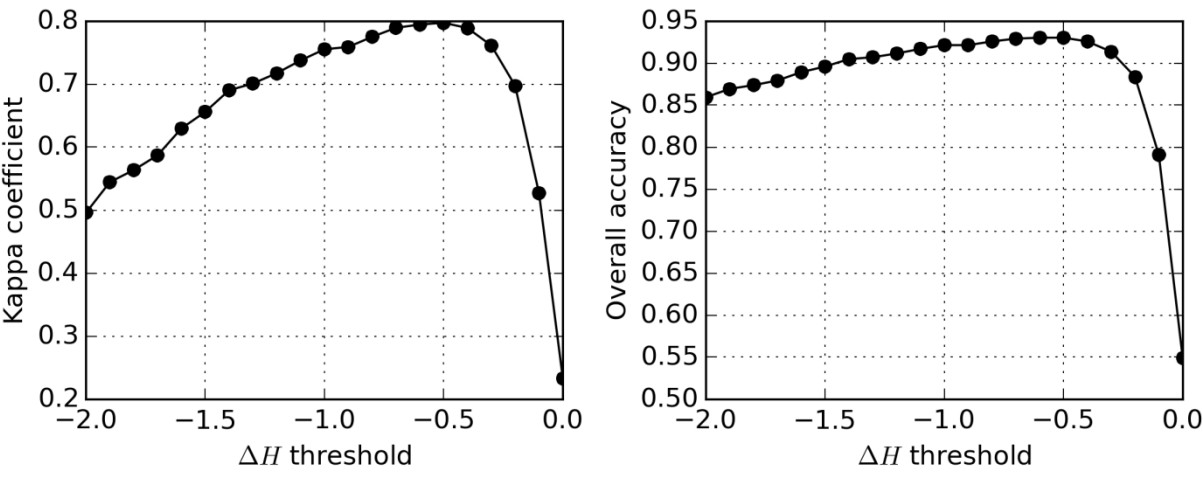

**Figure 9: Kappa coefficient (left) and overall accuracy (right) obtained from the comparison between the data surveyed by Yamada et al. (2017) and the estimated collapsed buildings based on different $\Delta H$ threshold values.**

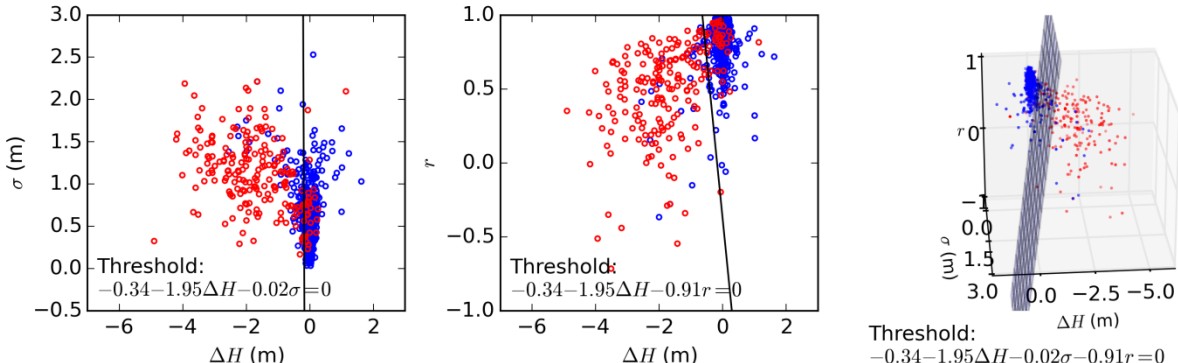

**Figure 10: Classification of collapsed (red) and non-collapsed (blue) buildings using the three parameters based on SVM.**

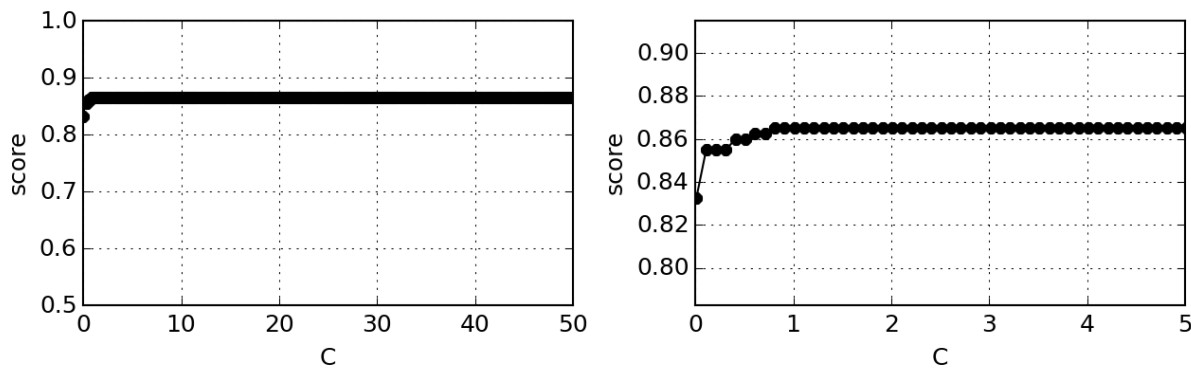

**Figure 11: Classifier's cross-validation accuracy as a function of C. Left: Overall evaluated range of C. Right: A closer look of values lower than 5.**

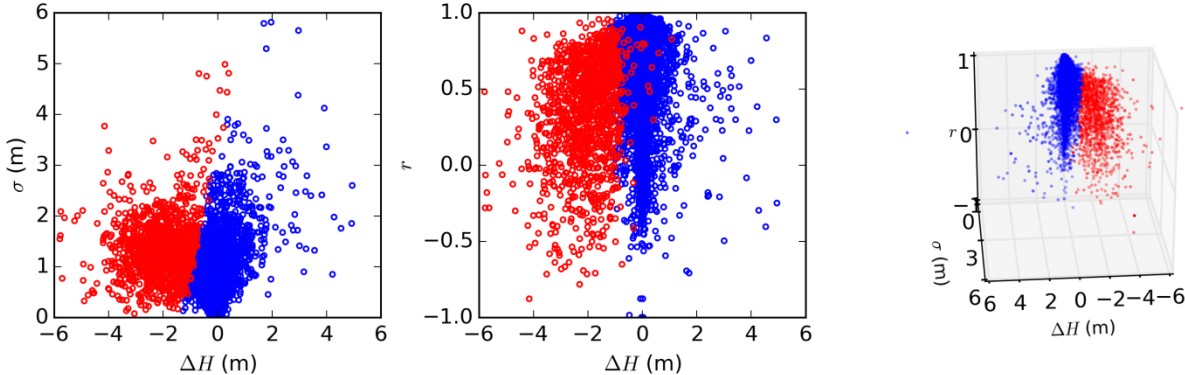

10      **Figure 12: Classification of collapsed (red) and non-collapsed (blue) buildings using the three parameters based on the K-means clustering method**

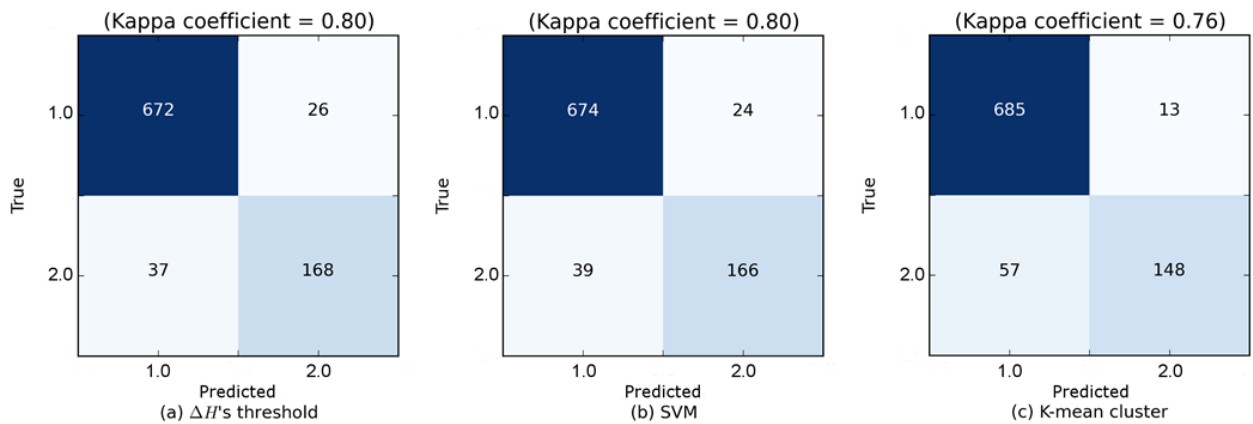

**Figure 13: Confusion matrix calculated from the comparison of the ground truth data and the predicted damage levels based on the $\Delta H$ threshold (a), SVM (b), and K-means clustering (c). Two damage levels, non-collapsed (1.0) and collapsed (2.0), were employed.**

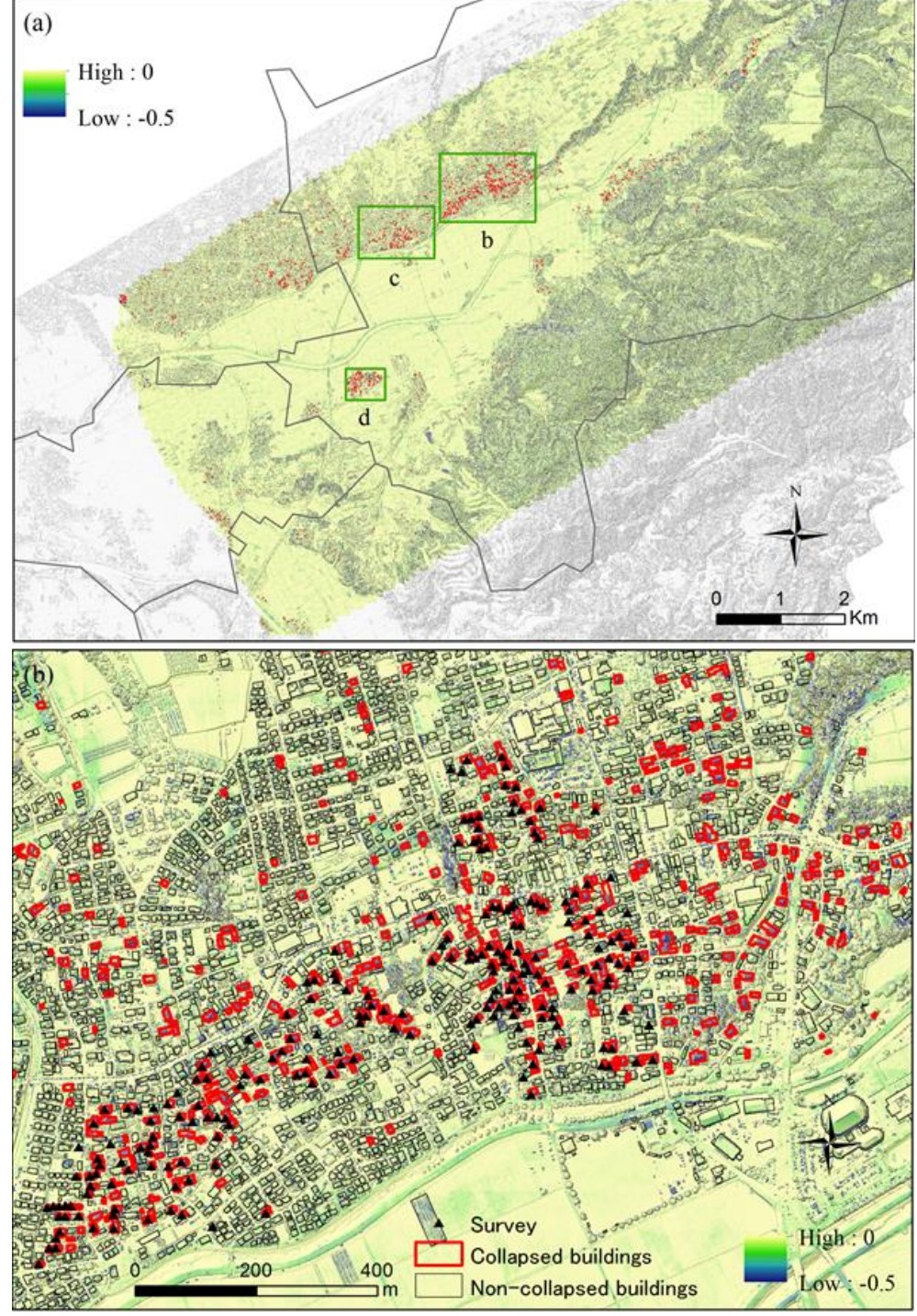

**Figure 14: (a) A map showing the distribution of collapsed ($\Delta H \leq$ -0.5 m) buildings, shown as red polygons, in the study area. The pixel color represents the difference in elevations between the BDSM and ADSM. The green squares show the locations of areas shown in Figure 13 (b) and Figure 14. (b) Close-up view of Area (b) where the collapsed buildings were concentrated. The black triangles show the D5 buildings from Yamada et al. (2017).**

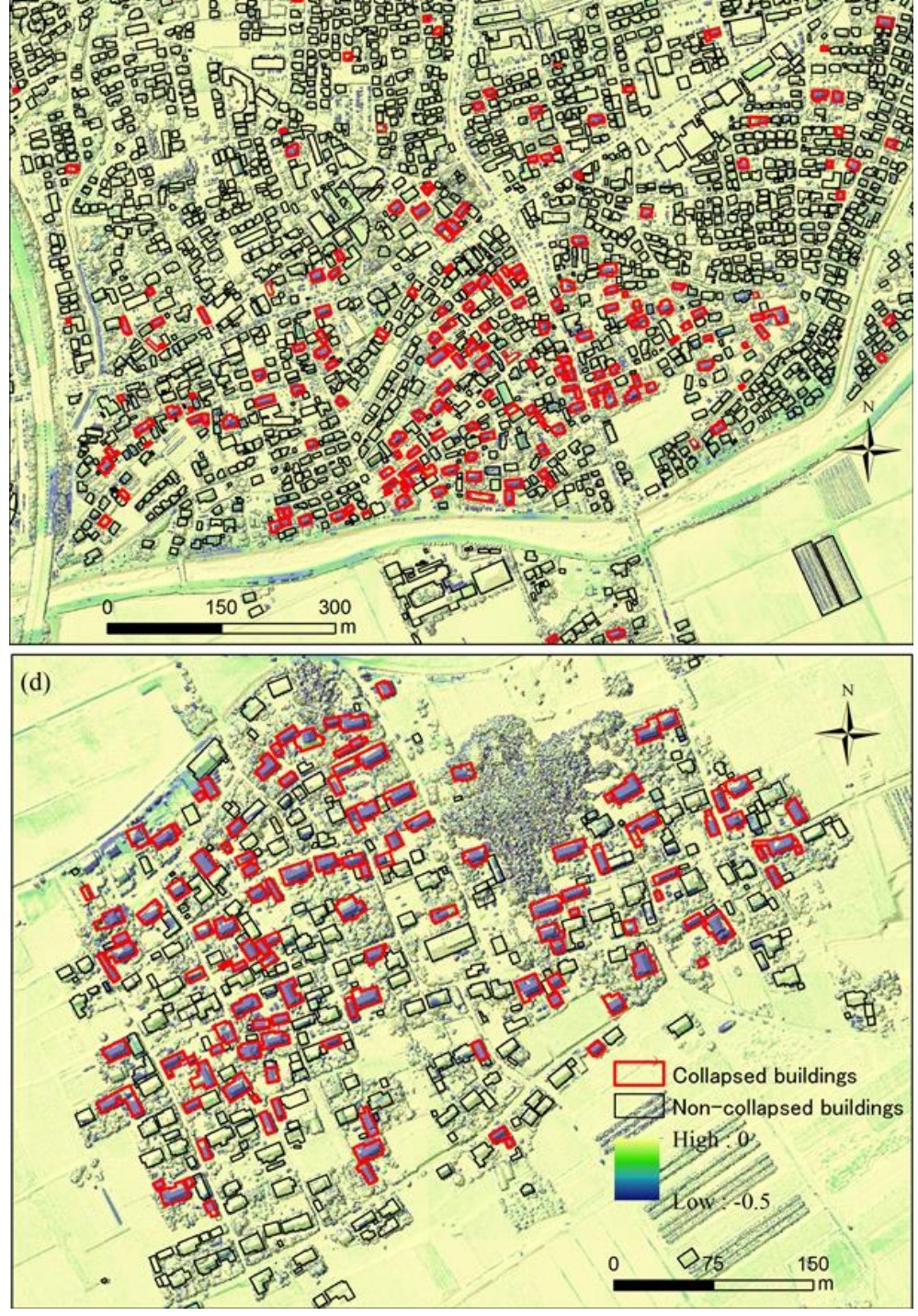

**Figure 15: Close-up view of Areas (c) and (d) in Figure 13 (a) where collapsed buildings are concentrated. The red and green polygons are the collapsed and non-collapsed buildings estimated using the threshold ($\Delta H \leq$ -0.5 m).**

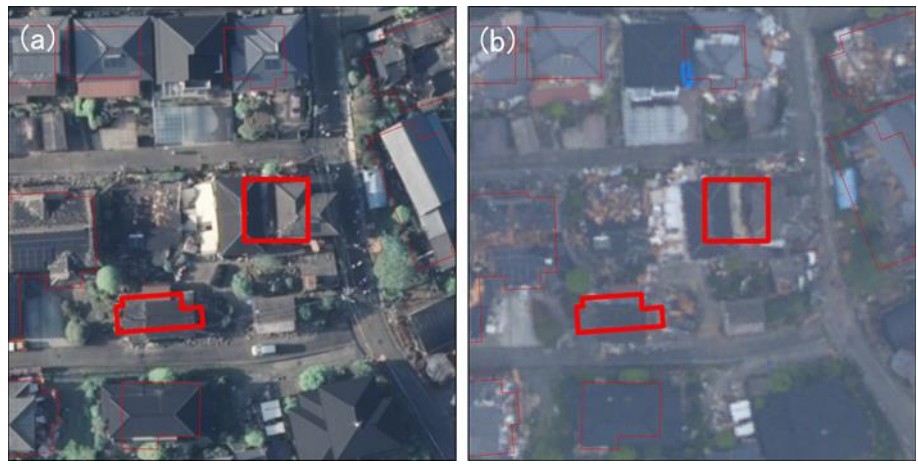

**Figure 16: (a) Aerial image taken on April 15; (b) Aerial image taken on April 23. The thick red polygons shows buildings that collapsed after the foreshock and were detected using the $\Delta H$ threshold.**