# Peer review of "Detection of collapsed buildings due to the 2016 Kumamoto, Japan, earthquake from LiDAR data"

_Natural Hazards and Earth System Sciences, 2017_

## Referee Comment (RC1) · Anonymous Referee #1 · 12 Jul 2017

General comment: This paper discusses the possibility of pre- and post-event lidar data in detecting collapsed buildings by an earthquake. Background and literature reviews were well described. Their conclusions are very clear and useful but may sound to be simple because only the difference of elevation is effective in identifying collapsed buildings. However, the paper quantitatively discusses the accuracy of other features such as standard deviation and correlation coefficient in the building damage detection. The conclusions are carefully verified by these multiple feature-based approach. Therefore, the reviewer judged that the paper deserves to be accepted. Then, some questions for better understanding are listed in the following specific comments.

Specific comment: Line 14 in Page 3 The post-event DSM is shifted to match the pre-event DSM by giving the permanent displacement. The displacements are given to

every pixels with the resolution of 50cm? In your previous paper (Moya et al. 2017), the displacement is calculated by 100m-grid size. How did you distribute the 100m-grid displacements to every 50cm pixels?

Line 27 in Page 3 In this study, building polygon is reduced by 1m in order to avoid the errors in matching of ADSM and BDSM. In the right column of Figures 4, the scale of the figures are not included. The reviewer could not judge the effect of the 1m-reduction of the building polygon and recommend to add not only the scales but also the 1m-reduced building polygons by dotted lines.

Equation (4) to (9) and Figure 12 In the equation (4) to (9), the subscript number 1 and 2 indicate non-collapsed and collapsed building, respectively. On the other hand, in Figure 12 the index number of the confusion matrix 0.0 and 1.0 indicate non-collapsed and collapsed building, respectively. The subscript number in the equations and Figure 12 should be unified for clearer understanding.

Line 1 in Page 7 The authors found that the K-means clustering provided lower accuracy but did not described the reason. Please describe the reason why the K-means clustering gives lower accuracy than SVM.

---

## Author Comment (AC1) · 30 Jul 2017

We acknowledge the referee for the insightful review. The comments have been considered in order to improve our manuscript. The details are addressed below. Please note we attach the new version of the manuscript, which might be subjected to further modifications after the other referee's comments.

COMMENT:

Line 14 in Page 3 The post-event DSM is shifted to match the pre-event DSM by giving the permanent displacement. The displacements are given to every pixels with the resolution of 50cm? In your previous paper (Moya et al. 2017), the displacement is calculated by 100m-grid size. How did you distributed the 100m-grid displacement to

every 50cm pixels?

Author's response:

The referee is right to point out we did not mention how we distributed the 100m-grid size displacement. We simply applied the displacement to all pixels inside the 100m-grid size. The authors decide it was enough considering that the displacement varies smoothly, as can be observed in Figure 2.

Author's change in manuscript:

The additional information is added in Line 16, Page 3 (See also attached file with the new version of the manuscript): "The permanent ground displacement was calculated by 100m-grid size, which is applied to the ADSM pixels within the grid-size. Figure 2 illustrates the calculated..."

COMMENT:

Line 27 in Page 3 In this study, building polygon is reduced by 1m in order to avoid the errors in matching of ADSM and BDSM. In the right column of Figures 4, the scale of the figures are not included. The reviewer could not judge the effect of the 1m reduction of the building polygon and recommend to add not only the scales but also the 1m-reduced building polygons by dotted lines.

Author's response:

In accordance with the referee's comment, we have modified Figure 4.

Author's change in manuscript:

The modified Figure 4 is observed in Page 13. Besides, the figure is now referenced in Line 29 Page 3: "For this reason, the building footprints were reduced by 1 m (i.e., the reduced polygon is located inside a building footprint), and they were projected onto the same reference system as that of the DSMs (Figure 4)"

COMMENT:

Equation (4) to (9) and Figure 12 In the equation (4) to (9), the subscript number 1 and 2 indicate non-collapsed and collapsed building, respectively. On the other hand, in Figure 12 the index number of the confusion matrix 0.0 and 1.0 indicate non-collapsed and collapsed building, respectively. The subscript number in the equations and Figure 12 should be unified for clearer understanding.

Author's response:

In accordance with the referee's comment, we have modified Figure 12.

Author's change in manuscript:

The modified figure 12 is located in page 17.

COMMENT:

Line 1 in Page 7 The authors found that the K-means clustering provided lower accuracy but did not described the reason. Please describe the reason why the K-means clustering gives lower accuracy than SVM.

Author's response:

The reason of differences in the accuracy rely on the nature of the methods. As it was mentioned. The SVM uses training data to calibrate the classification, while the K-means clustering does not use any training data. K-mean cluster considers only the distribution of the all data to perform the clasification.

Author's change in manuscript:

Additional comments are added in Line 31 Page 7: "Of the three methods evaluated here, K-means clustering exhibits the lowest accuracy. The main reason is that , unlike SVM method, K-means clustering does not use any training data. However, it produced a Kappa coefficient of 0.76 and an overall accuracy of 92%, which is still

quite good. The K-means clustering method can be useful for taking a first glance at the distribution of collapsed buildings because the method does not require any training data. The procedure is well-known and robust, with several efficient algorithms with proven fast convergence."

Please also note the supplement to this comment:
https://www.nat-hazards-earth-syst-sci-discuss.net/nhess-2017-186/nhess-2017-186-AC1-supplement.pdf

**Supplement:**

[revised manuscript text omitted]

---

## Short Comment (SC1) · 31 Jul 2017

In this paper, the authors provide a building damage detection approach based on pre- and post-event Digital Surface Models (DSMs) extracted from Light Detection and Ranging (LiDAR) instrument. They extracted three features from DSMs. Then the SVM classifier was employed to detect damaged buildings from the extracted features. The performance of SVM and K-Means clustering was compared with respect to each other. Topic is interesting and the study is valuable. As the authors mentioned, there are a little studies in this field. I would like to mention some issues that can improve the manuscript from my viewpoint: -Abstract: I think the first sentence is not necessary. -Abstract: "Different methods for extracting the collapsed . . . " please revise this sentence. -Introduction: The authors can use the following papers in the literature

review to improve it: ~Rehor, Miriam, et al. "Contribution of two plane detection algorithms to recognition of intact and damaged buildings in lidar data." The Photogrammetric Record 23.124 (2008): 441-456. ~Schweier, Christine, and Michael Markus. "Classification of collapsed buildings for fast damage and loss assessment." Bulletin of earthquake engineering 4.2 (2006): 177-192. -Page 2, Lines 26-31: the presented aim is not clear. -Page 3: "(i.e., the reduced polygon is located inside a building footprint)" there is no need to use parenthesis. -Page 4: BDSM?? ADSM?? -I think it is possible to present "Detection of damaged buildings" section in a better and logic manner. For example, they firstly provided accuracy assessment measures and then presented SVM method. Their positions can be changed. -Although SVM is a famous classifier, it is necessary to provide some descriptions about that since it is directly used in the methodology. -Please express parameters selected for implementing SVM and K-Means over the study area. How could you adjust their parameters? -Conclusion: Please provide some future studies.

---

## Author Comment (AC2) · 6 Aug 2017

We express our gratitude to Mr. Milad Janalipour for his comments. Please kindly find our response below. The updated version of the manuscript, after considering your comments, is attached.

(1) COMMENT

-Abstract: I think the first sentence is not necessary

AUTHOR'S RESPONSE

As mentioned in the paper, having a pair of Lidar data, before and after the earthquake, is not often. A pre-event Lidar data is available here because the strong foreshock (April

14). That is, the first mission was sent to record the effects of the foreshock. Therefore, we believe the first sentence is important to gives an overall view to the readers.

CHANGE IN MANUSCRIPT

No changes

(2) COMMENT -Abstract: "Different methods for extracting the collapsed . . ." please revise this sentence

AUTHOR'S RESPONSE

Following the comment, the sentence has been modified to: "Different methods were evaluated to extract collapsed building from the DSMs. The change . . ."

CHANGE IN MANUSCRIPT

The modified sentence is located at Line 13 Page 1.

(3) COMMENT

-Introduction: The authors can use the following papers in the literature review to improve it: ∼Rehor, Miriam, et al. "Contribution of two plane detection algorithms to recognition of intact and damaged buildings in lidar data." The Photogrammetric Record 23.124 (2008): 441-456. âĹijSchweier, Christine, and Michael Markus. "Classification of collapsed buildings for fast damage and loss assessment." Bulletin of earthquake engineering 4.2 (2006): 177-192.

AUTHOR'S RESPONSE

As suggested by the reviewer, we have included the mentioned papers.

CHANGE IN MANUSCRIPT

The paper of Schweier and Markus (2006) is referred at Page 2 Line 8-10: "Schweier and Markus (2006) pointed out Lidar data can be used to classify collapsed buildings. Thus, they proposed a modification of previous damage classification types (Okada

and Takai, 2000) and suggested features that can be extracted from Lidar data to classify collapsed buildings. However, applications of the framework were not provided."

The paper of Rehor et al. (2008) is refereed at Page 2 Line 14-16: "Rehor et al. (2008) proposed the use of a planes-based segmentation method to detect damaged buildings, where the number of unsegmented pixels in damaged buildings is larger than in undamaged buildings."

(4) COMMENT

-Page 2, Lines 26-31: the presented aim is not clear.

AUTHOR'S RESPONSE

The sentences have been rephrased to; "Therefore, this study explores the potential use of Lidar data to extract damaged buildings over the affected area. The difference of elevation, the standard deviation and the correlation coefficient were tested to obtain information regarding the damage state of buildings."

CHANGE IN MANUSCRIPT

The modified sentences begins at Page 2 Line 33.

(5) COMMENT

-Page 3: "(i.e., the reduced polygon is located inside a building footprint)" there is no need to use parenthesis.

AUTHOR'S COMMENT

With all due respect, there is not infraction on the use of parenthesis. It is only a matter of writing style.

CHANGE IN MANUSCRIPT

No changes.

(6) COMMENT

-Page 4: BDSM?? ADSM??

AUTHOR'S COMMENT

BDSM and ADSM represent the pre-event and post post-event digital surface models. The terms were introduced for the sake of brevity at Page 3 Line 8 of the original Discussion Manuscript.

CHANGE IN MANUSCRIPT

No changes

(7) COMMENT

-I think it is possible to present "Detection of damaged buildings" section in a better and logic manner. For example, they firstly provided accuracy assessment measures and then presented SVM method. Their positions can be changed.

AUTHOR'S COMMENT

One of the first steps on classification techniques is to evaluate the input features. In this case: $\Delta H$, $\sigma$, and r. I believe it is important first to evaluate their level of uncertainties before applying any classification method. Thus, we first evaluated our features. Then we applied some classification methods. Finally, we evaluated the accuracy of the classification. With all due respect, I do not think it is logic to switch the order.

CHANGE IN MANUSCRIPT

No changes

(8) COMMENT

-Although SVM is a famous classifier, it is necessary to provide some descriptions about that since it is directly used in the methodology.

AUTHOR'S COMMENT

Few comments regarding on the methodology are located at Page 6 Lines 9-13 of the original discussion manuscript.

CHANGE IN MANUSCRIPT

No changes

(9) COMMENT

-Please express parameters selected for implementing SVM and K-Means over the study area. How could you adjust their parameters?

AUTHOR'S COMMENT

Following the reviewer's suggestion, we have included some words regarding to the classification methods. For the case of k-means cluster, the only setting possible to manipulate is the initial values of the centers of each class. Here, we used a suggested k-means++ procedure to set the initial values, which makes the centers to be distant from each other. For the SVM classifier, considering we used a linear kernel, there is only one parameter that must be evaluated. We performed a cross-validation analysis for that purpose and concluded that there is not effect of the C parameter on our results. The result of the cross-validation is shown in a new figure (Figure 11)

CHANGE IN MANUSCRIPT

The information regarding k-means cluster is located at Page 7 Line 9: "The result is highly dependent on the initialization of the centroids. Here, k-means++ initialization scheme was used. K-means++ initializes the centroids to be distant from each other (Scikit-learn, 2017b)."

The information regarding SVM is located at Page 6 Line 19: "For a linear kernel SVM, the parameter C is the only value that must be considered. The parameter C trades off misclassification of training examples against simplicity of the decision surface. A

low C value makes the decision surface smooth and a high C value aims at classifying all training examples correctly (Skit-learn, 2017a). In this study a value C equals to 1 was used. In order to evaluate its effects, a cross-validation procedure was performed. Here, a range of C values are evaluated. For each C value, 80% of the surveyed data is selected randomly and used to calibrate the SVM classifier. The rest of the surveyed data is used to calculate a score that represents the accuracy. The overall accuracy was chose as the score. The procedure is repeated 5 times and the average is stored. Figure 11 shows the cross-validation accuracy. It is observed the accuracy remains mainly constant with small fluctuations at lower values. However, a difference of approximately 3% is observed between the worst and the best accuracy. Therefore, it is concluded that the C value did not affect the SVM classifier in our study"

(10) COMMENT

-Conclusion: Please provide some future studies.

AUTHOR'S COMMENT

As suggested by the reviewer, a future study is included.

CHANGE IN MANUSCRIPT

Page 8 Line 16: "It is expected that the use of Lidar data to extract damage areas will eventually increase in the near future. However, because of the current lack of data, the extraction of collapsed building using only post-event Lidar data will be addressed in a further publication."

Please also note the supplement to this comment:
https://www.nat-hazards-earth-syst-sci-discuss.net/nhess-2017-186/nhess-2017-186-AC2-supplement.pdf

---

## Referee Comment (RC2) · Anonymous Referee #2 · 9 Aug 2017

The paper examines the usage of LiDAR scans in detecting and characterizing collapsed building. General detection of buildings is based on geocoded database coupled with pre-event laser scanned data. Each building is then characterized by three parameters based on pre- and post-event scans: height differences ($\Delta$H), standard deviation ($\sigma$) and correlation (r). Based on these parameters the authors test three different methods to classify collapsed and non-collapsed building. They make further use of the parameters in order to characterize the collapse pattern. The paper is well written, and provides an approach to damaged building detection. However, some points should be considered: The detection of collapsed buildings is essentially change detection via laser scanning in urban areas. Nevertheless, the authors did not refer to such (or other) change-detection related works, where height difference is used as the

most reliable and efficient way to detect changes. It is unclear why this case should be different, yet it the authors consider it as an innovation. Although a major part of the paper deals with the classification of the collapsed buildings, this objective is not spelled out, and no reference to it – or to its importance – is made before page 4, where it is somewhat hidden within the general methodology of detecting collapsed buildings in general. The standard deviation and the correlation coefficient parameters hardly affected the detection, but were vital to identifying the pattern of the collapse. Though this is an interesting and new usage in these parameters, it is missing throughout the paper, especially in the discussion. As the discussion is quite short I would consider merging it with the conclusion to one "Discussion and conclusions" section.

Focused comments: Throughout the paper: please change "Lidar" to "LiDAR" Page 3, line 24: "geocoded building footprint dataset" – is that a vector map of the area? Line 27: "reduced by 1 m" – what does this mean? Is that an offset from the building boundaries? Page 6, Eq. 4-9: the order is reversed to the comments below. It should start with the elements that build the final coefficient, and not the other way around.

---

## Author Comment (AC3) · 20 Aug 2017

We acknowledge Referee #2 for his insightful comments. Please kindly find our response below and the updated version of the manuscript as an attached file. Besides, within the updated file, all the changes are in green color.

GENERAL WORDS:

The paper examines the usage of LiDAR scans in detecting and characterizing collapsed building. General detection of buildings is based on geocoded database coupled with pre-event laser scanned data. Each building is then characterized by three parameters based on pre- and post-event scans: height differences ($\Delta H$), standard deviation ($\sigma$) and correlation (r). Based on these parameters the authors test three

different methods to classify collapsed and non-collapsed building. They make further use of the parameters in order to characterize the collapse pattern. The paper is well written, and provides an approach to damaged building detection. However, some points should be considered:

(1) SPECIFIC COMMENT:

The detection of collapsed buildings is essentially change detection via laser scanning in urban areas. Nevertheless, the authors did not refer to such (or other) change-detection related works, where height difference is used as the most reliable and efficient way to detect changes. It is unclear why this case should be different, yet it the authors consider it as an innovation.

AUTHOR'S RESPONSE:

The referee is right to point out the lack of previous works related to change detection via laser scanning. It is because there are no publications related to change detection using laser scanning, as we mentioned in the introduction (Page 2 Line 12 of the updated manuscript). The main reason relies on the absence of a pre-event LiDAR data. However, from the references mentioned in the Introduction section. Maruyama et al. (2014) used different of height between DSMs obtained from aerial images. Furthermore, the lack of previous research on this subject highlights our desire to contribute in the use of a pair of Lidar data to extract collapsed building. Our results are supported by surveyed data, which was used as ground truth database.

CHANGE IN MANUSCRIPT:

In the reference of the paper of Maruyama et al. (2014) we modified to highlight that they used differences of elevation to detect collapsed buildings. See Page 2 Line20: "Instead of LiDAR data, Maruyama et al. (2014) constructed two digital surface models (DSMs) from two sets of aerial images: before and after the earthquake. Then, the collapsed buildings after the 2007 Niigata-Chuetsu-Oki, Japan, earthquake were

extracted using the difference of elevation between the DSMs."

(2) SPECIFIC COMMENT:

Although a major part of the paper deals with the classification of the collapsed buildings, this objective is not spelled out, and no reference to it – or to its importance – is made before page 4, where it is somewhat hidden within the general methodology of detecting collapsed buildings in general.

AUTHOR'S RESPONSE:

Indeed the referee is right. The main target is not clear in the firsts sections. We acknowledge the referee for highlighting this issue. Perhaps the current version highlights the time-line of our research on this topic. Where, at the early stage we intended to extract several levels of damage. However, at the end, only collapsed buildings were extracted with high accuracy. In the new version of the manuscript, we clarify that the major part of this manuscript deals with the extraction of collapsed buildings.

CHANGE IN MANUSCRIPT:

Page 2, Line 34: "Therefore, this study explores the potential use of Lidar data to extract collapsed buildings over the affected area. The difference of elevation, the standard deviation and the correlation coefficient were tested for this purpose."

(3) SPECIFIC COMMENT:

The standard deviation and the correlation coefficient parameters hardly affected the detection, but were vital to identifying the pattern of the collapse. Though this is an interesting and new usage in these parameters, it is missing throughout the paper, especially in the discussion

AUTHOR'S RESPONSE:

The collapsed patterns are correlated to the failure mechanism of the buildings, which is important for forensic engineering, investigation of failures, etc. We have included

these comments in the discussion.

CHANGE IN MANUSCRIPT:

Page 7 Line 32: "The collapsed patterns are correlated to the failure mechanism in buildings, which might highlight some deficiencies of the design codes that was used during the construction process. A detailed understanding of failure mechanism is important to the practice of forensic engineering, the investigation of failures and other performance problems. Moreover, with further evaluation, the collapsed pattern can contribute to future improvements of the construction design codes. Unfortunately, it was not possible to calibrate a threshold that can properly classify different collapsed patterns. The main reason is because there was not information related to collapse patterns in the surveyed data. Perhaps this task can be done in further research after new surveyed data are released."

(4)SPECIFIC COMMENT:

As the discussion is quite short I would consider merging it with the conclusion to one "Discussion and conclusions" section.

AUTHOR'S RESPONSE:

The discussion section has been enlarged after (1) including a discussion related to collapsed pattern, which was the previous comment and (2) including some word related to future studies, which was a suggestion from the Interactive Discussion Forum (Page 8 Line 22).

CHANGE IN MANUSCRIPT:

No changes

FOCUSED COMMENTS

(4) SPECIFIC COMMENT:

Throughout the paper: please change "Lidar" to "LiDAR".

AUTHOR'S RESPONSE:

As suggested by the reviewer, Lidar has been changed to LiDAR throughout the paper.

CHANGE IN MANUSCRIPT:

Throughout the paper (please see attached file)

(5) SPECIFIC COMMENT:

Page 3, line 24: "geocoded building footprint dataset" – is that a vector map of the area?

AUTHOR'S RESPONSE:

The referee is right to point out the technical word is vector data. However, we decide not to use it because NHESS gathers readers from different disciplines that might not be familiar with GIS technical words.

CHANGE IN MANUSCRIPT:

No changes

(6) SPECIFIC COMMENT:

Line 27: "reduced by 1 m" – what does this mean? Is that an offset from the building boundaries?

AUTHOR'S RESPONSE:

That is right, to make it clear and following the comment of referee #1, the reduced building boundary is now included in Figure 4 as dashed lines.

CHANGE IN MANUSCRIPT:

The updated figure 4 is located in Page 14 and referred in Page 4 Line 4 (See attached

file).

(7) SPECIFIC COMMENT:

Page 6, Eq. 4-9: the order is reversed to the comments below. It should start with the elements that build the final coefficient, and not the other way around.

AUTHOR'S RESPONSE:

As suggested by the referee, the order of the equations has been modified.

CHANGE IN MANUSCRIPT:

Page 6, Eq. 4-9 (See attached file).

Please also note the supplement to this comment:
https://www.nat-hazards-earth-syst-sci-discuss.net/nhess-2017-186/nhess-2017-186-AC3-supplement.pdf

─────────────────────────────

---

## Author Response (AR1)

**RESPONSE TO THE EDITOR**

**General words:**

Dear Professor Luis Moya,

I have got two reviewer reports for your manuscript: 'Detection of collapsed buildings due to the 2016 Kumamoto, Japan, earthquake from Lidar data'. Both reviewers concluded that the paper has scientific values, but included important comments which I found critical.

**Specific comments**

**Comment:**

In order to make the paper publishable please revise it, following carefully the comments and suggestions made by both reviewers.

**Author's response**

As suggested by the editor, we have followed carefully the reviewer reports and update our manuscript according to them.

**Change in manuscript**

Details of the reviewer comments together with our response and changes in the manuscript are presenter further in this document. All the changes are marked as blue in the attached new manuscript.

**Comment:**

In general, I suggest to increase the scientific discussion volume over the technical one. Also relate to the EQ events, and discuss if the number of collapsed building is similar to other cases of M=7 earthquakes, or the fact that it was a sequence of M=6.2 followed by M=7 event affected the results.

**Author's response:**

We have discussed whether is possible to make a proper comparison with other event of M=7 regarding to the number of collapsed buildings. There are several factors that must be included, such as intensity of the strong motion, density of buildings, performance of buildings, etc. Therefore, it is difficult to perform a comparison of events with same magnitude. Regarding to the effect of the first event of M=6.2, conclusions from numerical models of wood-frame buildings are summarized.

**Change in manuscript:**

The following paragraphs have been included in the discussion section (Page 8, line 26):

   "*Finally we would like to discuss on the building damage rate in the Kumamoto earthquake, compared with other recent M7-level crustal earthquakes in Japan. In the 1995 Kobe earthquake (Mw 6.9), about 49 thousand buildings were collapsed (G5 in EMS-98 scale) or severely damaged (G4) out of 560 thousand buildings in the affected urban area (Building Research Institute, 1996). The recorded strong*

*motion distribution in the Kobe earthquake was in the similar level as that of the Kumamoto earthquake (Yamaguchi and Yamazaki, 2001). But in 1995, the number of strong motion accelerometers was much less, about only ten in the hard-hit zone. The built-up density of the affected area was much higher in the Kobe region than that of the Kumamoto region. Considering these differences, it is difficult to conclude which earthquake was more destructive. From our experiences (Yamazaki and Murao, 2000; Yamaguchi and Yamazaki, 2000), the severity of damage for wood-frame houses in Mashiki town was in the similar level as that of the hard-hit zone of the Kobe region. There were a few more recent M7-level crustal earthquakes in Japan, such as the 2004 Niigata-Chuetsu earthquake (Mw 6.6) and the 2007 Niigata-Chuetsu-Oki earthquake (Mw 6.6). But the population density of the affected urban areas was fewer in these events, and thus it is again difficult to compare their damage situations (Nagao et al., 2011) with that of Kumamoto although their strong motion levels were again comparable with that of the Kumamoto event.*

*Another point of discussion is whether or not the Mw 6.2 foreshock influenced the overall damage situation of Mashiki town. Our preliminary conclusion is "partially yes, but not so much". Based on numerical analyses on the collapse behaviour of typical wood-frame buildings (Building Research Institute, 2015), those built by the old seismic-code (before 1982) were mostly collapsed only by the main-shock's excitation (Haya et al., 2016), even without pre-shaken by the foreshock (Suto et al., 2017). But in some cases, building models were collapsed only under the sequence of the foreshock and main-shock excitations. More detailed results on this matter will be presented in the near future."*

**RESPONSE TO THE REFEREE #1**

**Comment**

Line 14 in Page 3 The post-event DSM is shifted to match the pre-event DSM by giving the permanent displacement. The displacements are given to every pixels with the resolution of 50cm? In your previous paper (Moya et al. 2017), the displacement is calculated by 100m-grid size. How did you distributed the 100m-grid displacement to every 50cm pixels?

**Author's response:**

The referee is right to point out we did not mention how we distributed the 100m-grid size displacement. We simply applied the displacement to all pixels inside the 100mgrid size. The authors decide it was enough considering that the displacement varies smoothly, as can be observed in Figure 2.

**Change in manuscript:**

The additional information is added in Page 3, line 19:

*"The permanent ground displacement was calculated by 100m-grid size, and then it is applied to the ADSM pixels within the grid-size."*

**Comment**

Line 27 in Page 3 In this study, building polygon is reduced by 1m in order to avoid the errors in matching of ADSM and BDSM. In the right column of Figures 4, the scale of the figures are not included. The reviewer could not judge the effect of the 1m reduction of the building polygon and recommend to add not only the scales but also the 1m-reduced building polygons by dotted lines.

**Author's response:**

In accordance with the referee's comment, we have modified Figure 4.

**Change in manuscript:**

The modified Figure 4 is observed in Page 15. Besides, the figure is now referenced in Page 4, line 2.

**Comment**

Equation (4) to (9) and Figure 12 In the equation (4) to (9), the subscript number 1 and 2 indicate non-collapsed and collapsed building, respectively. On the other hand, in Figure 12 the index number of the confusion matrix 0.0 and 1.0 indicate non-collapsed and collapsed building, respectively. The subscript number in the equations and Figure 12 should be unified for clearer understanding.

**Author's response:**

In accordance with the referee's comment, we have modified Figure 12.

**Change in manuscript:**

Because other comments, the Figure 12 is now Figure 13 and is located in Page 20.

**Comment**

Line 1 in Page 7 The authors found that the K-means clustering provided lower accuracy but did not described the reason. Please describe the reason why the K-means clustering gives lower accuracy than SVM.

**Author's response:**

The reason of differences in the accuracy relies on the nature of the methods. As it was mentioned. The SVM uses training data to calibrate the classification, while the K-means clustering does not use any training data. K-mean cluster considers only the distribution of the all data to perform the classification.

**Change in manuscript:**

Additional comments are added in page 8, line 21:

"*Of the three methods evaluated here, K-means clustering exhibits the lowest accuracy. The main reason is that, unlike SVM method, K-means clustering does not use any truth data. However, it produced a Kappa coefficient of 0.76 and an overall accuracy of 92%, which is still quite good. The K-means clustering method can be useful for taking a first glance at the distribution of collapsed buildings because the method does not require any training data. The procedure is well-known and robust, with several efficient algorithms with proven fast convergence.*"

**RESPONSE TO THE REFEREE #2**

**General words:**

The paper examines the usage of LiDAR scans in detecting and characterizing collapsed building. General detection of buildings is based on geocoded database coupled with pre-event laser scanned data. Each building is then characterized by three parameters based on pre- and post-event scans: height differences ($\Delta H$), standard deviation ($\sigma$) and correlation (r). Based on these parameters the authors test three C1 NHESSD Interactive comment Printer-friendly version Discussion paper different methods to classify collapsed and non-collapsed building. They make further use of the parameters in order to characterize the collapse pattern. The paper is well written, and provides an approach to damaged building detection. However, some points should be considered:

**Specific comments**

**Comment**

The detection of collapsed buildings is essentially change detection via laser scanning in urban areas. Nevertheless, the authors did not refer to such (or other) change detection related works, where height difference is used as the most reliable and efficient way to detect changes. It is unclear why this case should be different, yet it the authors consider it as an innovation.

**Author's response:**

The referee is right to point out the lack of previous works related to change detection via laser scanning. It is because there are no publications related to change detection using laser scanning, as we mentioned in the introduction (Page 2 Line 12 of the updated manuscript). The main reason relies on the absence of a pre-event LiDAR data. However, from the references mentioned in the Introduction section. Maruyama et al. (2014) used different of height between DSMs obtained from aerial images. Moreover, a new reference was included: Schweiver and Markus (2006). In their publication they pointed out the importance of usinf LiDAR data to classify collapsed buildings.

  Furthermore, the lack of previous research on this subject highlights our desire to contribute in the use of a pair of Lidar data to extract collapsed building. Regarding if the results are reliable and clear, our results are supported by surveyed data, which was used as ground truth database.

**Change in manuscript:**

In the sentence that refers to the paper of Maruyama et al. (2014), we modified it to highlight that they used differences of elevation to detect collapsed buildings. See Page 2 Line21:

  *"Instead of LiDAR data, Maruyama et al. (2014) constructed two digital surface models (DSMs) from two sets of aerial images: before and after the earthquake. Then, the collapsed buildings after the 2007 Niigata-Chuetsu-Oki, Japan, earthquake were extracted using the difference of elevation between the DSMs."*

The reference of the work of Schweier and Markus (2006) is located at page 2, line 9:

*"Schweier and Markus (2006) pointed out airborne light detection and ranging (LiDAR) data can be used*

*to classify collapsed buildings. Thus, they proposed a modification of previous damage classification types (Okada and Takai, 2000) and suggested features that can be extracted from LiDAR data to classify collapsed buildings. However, applications of the framework were not provided.*"

**Comment**

Although a major part of the paper deals with the classification of the collapsed buildings, this objective is not spelled out, and no reference to it – or to its importance – is made before page 4, where it is somewhat hidden within the general methodology of detecting collapsed buildings in general.

**Author's response:**

Indeed the referee is right. The main target is not clear in the firsts sections. We acknowledge the referee for highlighting this issue. Perhaps the current version highlights the time-line of our research on this topic. Where, at the early stage we intended to extract several levels of damage. However, at the end, only collapsed buildings were extracted with high accuracy. In the new version of the manuscript, we clarify in the introduction that the major part of this manuscript deals with the extraction of collapsed buildings.

**Change in manuscript:**

Page 3, line 1:

"*Therefore, this study explores the potential use of LiDAR data to extract collapsed buildings over the affected area. The difference of elevation, the standard deviation and the correlation coefficient were tested for this purpose. The detection of collapsed building is crucial because it produces secondary effects such as casualties and blockages to the road network.*"

**Comment**

The standard deviation and the correlation coefficient parameters hardly affected the detection, but were vital to identifying the pattern of the collapse. Though this is an interesting and new usage in these parameters, it is missing throughout the paper, especially in the discussion.

**Author's response:**

The collapsed patterns are correlated to the failure mechanism of the buildings, which is important for forensic engineering, investigation of failures, etc. We have included these comments in the discussion

**Change in manuscript:**

Page 8, line 4:

"*The collapsed patterns are correlated to the failure mechanism in buildings, which might highlight some deficiencies of the design codes that was used during the construction process. A detailed understanding of failure mechanism is important to the practice of forensic engineering, the investigation of failures and other performance problems. Moreover, with further evaluation, the collapsed pattern might contribute to future improvements of the construction design codes. Unfortunately, it was not possible to calibrate a threshold that can properly classify different collapsed patterns. The main reason is because there was*

*not information related to collapse patterns in the surveyed data. Perhaps this task can be done in further research after new surveyed data are released*"

**Comment**

As the discussion is quite short I would consider merging it with the conclusion to one "Discussion and conclusions" section.

**Author's response:**

The discussion section have been extended after considering the reviewer comments

**Change in manuscript:**

No changes

**Comment**

Throughout the paper: please change "Lidar" to "LiDAR".

**Author's response:**

As suggested by the reviewer, Lidar has been changed to LiDAR throughout the paper.

**Change in manuscript:**

Throughout the paper (please see the updated manuscript)

**Comment**

Page 3, line 24: "geocoded building footprint dataset" – is that a vector map of the area?

**Author's response:**

The referee is right to point out the technical word is vector data. However, we decide not to use it because NHESS gathers readers from different disciplines that might not be familiar with GIS technical words.

**Change in manuscript:**

No changes

**Comment**

Line 27: "reduced by 1 m" – what does this mean? Is that an offset from the building boundaries?

**Author's response:**

That is right, to make it clear and following the comment of referee #1, the reduced building boundary is now included in Figure 4 as dashed lines.

**Change in manuscript:**

The updated figure 4 is located in Page 15 and referred in Page 4 Line 2.

**Comment**

Page 6, Eq. 4-9: the order is reversed to the comments below. It should start with the elements that build the final coefficient, and not the other way around.

**Author's response:**

As suggested by the referee, the order of the equations has been modified.

**Change in manuscript:**

Page 6, Eq. 4-9 (See attached file).

**RESPONSE TO THE MILAD JANALIPOUR (INTERACTIVE DISCUSSION)**

**Comment:**

-Abstract: I think the first sentence is not necessary

**Author's response:**

As mentioned in the paper, having a pair of Lidar data, before and after the earthquake, is not often. A pre-event Lidar data is available here because the strong foreshock (April 14). That is, the first mission was sent to record the effects of the foreshock. Therefore, we believe the first sentence is important to gives an overall view to the readers.

**Change in manuscript:**

No changes

**Comment:**

-Abstract: "Different methods for extracting the collapsed . . ." please revise this sentence

**Author's response:**

Following the comment, the sentence has been modified to: "*Different methods were evaluated to extract collapsed building from the DSMs*".

**Change in manuscript:**

The modified sentence is located at Line 13 Page 1.

**Comment:**

-Introduction: The authors can use the following papers in the literature review to improve it: ~Rehor, Miriam, et al. "Contribution of two plane detection algorithms to recognition of intact and damaged buildings in lidar data." The Photogrammetric Record 23.124 (2008): 441-456. âLijSchweier, Christine, and Michael Markus. "Classification of collapsed buildings for fast damage and loss assessment." Bulletin of earthquake engineering 4.2 (2006): 177-192.

**Author's response:**

As suggested by the reviewer, we have included the mentioned papers.

**Change in manuscript:**

The paper of Schweier and Markus (2006) is referred at Page 2 Line 9:

"*Schweier and Markus (2006) pointed out Lidar data can be used to classify collapsed buildings. Thus, they proposed a modification of previous damage classification types (Okada and Takai, 2000) and suggested features that can be extracted from Lidar data to classify collapsed buildings. However, applications of the framework were not provided.*"

The paper of Rehor et al. (2008) is refereed at Page 2 Line 15:

"*Rehor et al. (2008) proposed the use of a planes-based segmentation method to detect damaged buildings, where the number of unsegmented pixels in damaged buildings is larger than in undamaged buildings.*"

**Comment**

-Page 2, Lines 26-31: the presented aim is not clear.

**Author's response:**

The sentences have been rephrased to;

"*Therefore, this study explores the potential use of LiDAR data to extract collapsed buildings over the affected area. The difference of elevation, the standard deviation and the correlation coefficient were tested for this purpose. The detection of collapsed building is crucial because it produces secondary effects such as casualties and blockages to the road network.*"

**Change in manuscript:**

The modified sentences begins at Page 3 Line 1.

**Comment:**

-Page 3: "(i.e., the reduced polygon is located inside a building footprint)" there is no need to use parenthesis.

**Author's response:**

With all due respect, there is not infraction on the use of parenthesis. It is only a matter of writing style.

**Change in manuscript:**

No changes

**Comment:**

-Page 4: BDSM?? ADSM??

**Author's response:**

BDSM and ADSM represent the pre-event and post post-event digital surface models. The terms were

introduced for the sake of brevity at Page 3 Line 8 of the original Discussion Manuscript.

**Change in manuscript:**

No changes

**Comment:**

-I think it is possible to present "Detection of damaged buildings" section in a better and logic manner. For example, they firstly provided accuracy assessment measures and then presented SVM method. Their positions can be changed.

**Author's response:**

One of the first steps on classification techniques is to evaluate the input features. In this case: $\Delta H$, $\sigma$, and r. I believe it is important first to evaluate their level of uncertainties before applying any classification method. Thus, we first evaluated our features. Then we applied some classification methods. Finally, we evaluated the accuracy of the classification. With all due respect, I do not think it is logic to switch the order.

**Change in manuscript:**

No changes

**Comment:**

-Although SVM is a famous classifier, it is necessary to provide some descriptions about that since it is directly used in the methodology.

**Author's response:**

Few comments regarding on the methodology are located at Page 6 Lines 12-16

**Change in manuscript:**

No changes

**Comment:**

-Please express parameters selected for implementing SVM and K-Means over the study area. How could you adjust their parameters?

**Author's response:**

Following the reviewer's suggestion, we have included some words regarding to the classification methods. For the case of k-means cluster, the only setting possible to manipulate is the initial values of the centers of each class. Here, we used a suggested k-means++ procedure to set the initial values, which makes the centers to be distant from each other. For the SVM classifier, considering we used a linear kernel, there is only one parameter that must be evaluated. We performed a cross-validation analysis for that purpose and concluded that there is not effect of the C parameter on our results. The result of the cross-validation is shown in a new figure (Figure 11)

**Change in manuscript:**

The information regarding k-means cluster is located at Page 7 Line 12:

  "*The result is highly dependent on the initialization of the centroids. Here, k-means++ initialization scheme was used. K-means++ initializes the centroids to be distant from each other (Scikit-learn, 2017b).*"

The information regarding SVM is located at Page 6 Line 22:

"*For a linear kernel SVM, a parameter C is the only value that must be considered. The parameter C trades off misclassification of training examples against simplicity of the decision surface. A low C value makes the decision surface smooth and a high C value aims at classifying all training examples correctly (Skit-learn, 2017a). In this study a value C equals to 1 was used. In order to evaluate its effects, a cross-validation procedure was performed. Here, a range of C values are evaluated. For each C value, 80% of the surveyed data is selected randomly and used to calibrate the SVM classifier. The rest of the surveyed data is used to calculate a score that represents the accuracy. The overall accuracy was chose as the score. The procedure is repeated 5 times and the average is stored. Figure 11 shows the cross-validation accuracy. It is observed the accuracy remains mainly constant with small fluctuations at lower values. However, a difference of approximately 3% is observed between the worst and the best accuracy. Therefore, it is concluded that the C value did not affect the SVM classifier in our study.*"

**Comment**

-Conclusion: Please provide some future studies.

**Author's response:**

As suggested by the reviewer, a future study is included.

**Change in manuscript:**

Page 9 Line 24:

  "*It is expected that the use of LiDAR data to extract damage areas will eventually increase in the near future. However, because of the current lack of data, the implementation of a method to extract of collapsed building using only post-event LiDAR data is important and will be considered in a future study. Additional future studies related to the use of these LiDAR data are the quantification of debris expansion that is related with road blockage, and the extraction of landslides.*"

---

## Referee Report (RR1)

The paper examines the usage of airborne LiDAR scans in detecting and characterizing collapsed buildings. As an initial phase, buildings are detected via geocoded database coupled with pre-event laser scanned data. Each building is then characterized by three parameters based on pre- and post-event scans: height differences ($\Delta H$), standard deviation ($\sigma$) and correlation ($r$). Based on these parameters the authors test three different methods to classify collapsed and non-collapsed buildings. They make further use of the parameters in order to characterize the collapse pattern.

The paper is well written, and changes based on previous comments were implemented within the text. However, some small details should be considered:

General comments:

- New paragraphs, which were added after revision, should be reviewed to improve language.

Focused comments:

Page 2, lines 8-12: it is unclear what exactly was suggested. Which features were extracted and what did they propose doing with them?

Page 3, lines 3-4: the sentence "The detection of collapsed building…" seems redundant, and is ill phrased.

Page 6, lines 22-25: I think that an equation will be of help here. It is unclear what the authors refer to in respect with parameter C, and if the purpose was to clarify SVM, it was missed here.

---

## Author Response (AR2)

**Second Response to Referee #2**

We acknowledge Referee #2 for his comments. Please kindly find our response below and the updated version of the manuscript as an attached file.

GENERAL COMMENTS:
New paragraphs, which were added after revision, should be reviewed to improve language:

AUTHOR'S RESPONSE:
As suggested by the referee, the new paragraphs were revised.

SPECIFIC COMMENT:
Page 2, lines 8-12: it is unclear what exactly was suggested. Which features were extracted and what did they propose doing with them?

AUTHOR'S RESPONSE:
We have extended the summary of Schweier and Markus (2006)'s paper. It includes now the proposed features and the purpose of their use.

CHANGE IN MANUSCRIPT:
See Page 2 Line 9-15 (text in red in the updated manuscript):
*"Schweier and Markus (2006) pointed out that airborne light detection and ranging (LiDAR) data can be used to classify collapsed buildings using the following geometrical features of a building extracted from LiDAR data: the height change from the initial one, the reduction of the total volume, the footprint borders, the inclination of the structure, the debris spread outside the footprint, the additional covered area outside the footprint, and the damage situation of the roof. They proposed a modification of the previous damage classification method (Okada and Takai, 2000) using these geometrical features. Although they suggested the use of airborne LiDAR data in extraction of collapsed buildings, applications to real cases were not provided."*

SPECIFIC COMMENT:
Page 3, lines 3-4: the sentence "The detection of collapsed building…" seems redundant, and is ill phrased.

AUTHOR'S RESPONSE

The mentioned sentence has been rephrased.

CHANGE IN MANUSCRIPT:

Page 3, Line 6-7 (text in red in the updated manuscript):

*"Building collapse is still the main cause of casualties and hence its prompt recognition is crucial for search and rescue operations."*

SPECIFIC COMMENT:

Page 6, lines 22-25: I think that an equation will be of help here. It is unclear what the authors refer to in respect with parameter C, and if the purpose was to clarify SVM, it was missed here.

AUTHOR'S RESPONSE

Following the referee's comment, a set of mathematical expressions were added to clarify from where the parameter C comes.

CHANGE IN MANUSCRIPT:

Page 7 Line 1-5 (text in red in the updated manuscript):

"For a linear kernel SVM, the vector $\mathbf{w}$ perpendicular to the decision plane is defined by the following expression:

$$\mathbf{w} = \sum_i \alpha_i y_i \mathbf{x}_i \tag{10}$$

where $\mathbf{x}_i$ is a training vector that contains the three parameters ($\Delta H$, $\sigma$ and $r$), $y_i$ represent the class that can be either 1 or -1, and the coefficients $\alpha_i$ are obtained by solving the following problem:

$$\min_{\boldsymbol{\alpha}} \left( \frac{1}{2} \boldsymbol{\alpha}^T \mathbf{Q} \boldsymbol{\alpha} - \mathbf{e}^T \boldsymbol{\alpha} \right) \tag{11}$$

$$Q_{ij} = y_i y_j \left( \mathbf{x}_i \cdot \mathbf{x}_j \right) \tag{12}$$

$$\le \alpha_i \le C, \, i = 1,\dots,n \tag{13}$$

where $e$ is a vector whose elements are all ones, C is the upper bound and is used as a regularization parameter."